# Towards Understanding the Role of Sharpness-Aware Minimization Algorithms for Out-of-Distribution Generalization

## Abstract

Recently, sharpness-aware minimization (SAM) has emerged as a promising method to improve generalization by minimizing sharpness, which is known to correlate well with generalization ability. Since the original proposal of SAM by Foret et al. (2021), many variants of SAM have been proposed to improve its accuracy and efficiency, but comparisons have mainly been restricted to the i.i.d. setting. In this paper we study SAM for out-of-distribution (OOD) generalization. First, we perform a comprehensive comparison of eight SAM variants on zero-shot OOD generalization, finding that the original SAM outperforms the Adam baseline by 4.76% and the strongest SAM variants outperform the Adam baseline by 8.01% on average. We then provide an OOD generalization bound in terms of sharpness for this setting. Next, we extend our study of SAM to the related setting of gradual domain adaptation (GDA), another form of OOD generalization where intermediate domains are constructed between the source and target domains, and iterative self-training is done on intermediate domains, to improve the overall target domain error. In this setting, our experimental results demonstrate that the original SAM outperforms the baseline of Adam on each of the experimental datasets by 0.82% on average and the strongest SAM variants outperform Adam by 1.52% on average. We then provide a generalization bound for SAM in the GDA setting. Asymptotically, this generalization bound is no better than the one for self-training in the literature of GDA. This highlights a further disconnection between the theoretical justification for SAM versus its empirical performance, as noted in Wen et al. (2023), which found that low sharpness alone does not account for all of SAM's generalization benefits. For future work, we provide several potential avenues for obtaining a tighter analysis for SAM in the OOD setting. In summary, our theoretical results provide a solid starting point for analyzing SAM in OOD settings, and our experimental results demonstrate that SAM can be applied to OOD settings to significantly improve accuracy, and that newer variants of SAM can be leveraged for further improvements in accuracy.

## 1 Introduction

A promising new optimization algorithm called Sharpness-Aware Minimization (SAM) exploits a known relationship between the "flatness" of a minimum and its i.i.d. generalization (Jiang et al., 2020; Dinh et al., 2017; Keskar et al., 2017; Hochreiter & Schmidhuber, 1997), proposing a robust optimization procedure that leads to significant performance gains in the i.i.d. setting (Foret et al., 2021). However, SAM remains understudied in the out-of-distribution (OOD) generalization setting, which is central topic of interest at this time, both theoretically and empirically (Kumar et al., 2020; Cha et al., 2021; Arjovsky et al., 2020; He et al., 2024; Ye et al., 2021; Zhao et al., 2022; 2019; Ben-David et al., 2010). Furthermore, a number of SAM variants have been proposed to improve the accuracy and efficiency of the original SAM algorithm, with comparisons mainly restricted to the i.i.d. setting (Kim et al., 2022; Li et al., 2024; Du et al., 2022; Kwon et al., 2021; Mueller et al., 2024; Liu et al., 2022; Ni et al., 2022).

Inspired by the practical success of SAM and its variants in the i.i.d. setting and interested in its potential to enhance OOD generalization, we perform an empirical and theoretical study of OOD generalization with SAM and its variants. Our work makes the following contributions:

1. In Section 2, we provide an introduction to eight SAM variants, including the original SAM. Then, in Section 3, we perform a comprehensive comparison of the zero-shot OOD generalization capabilities for these eight SAM variants. We find over four empirical benchmarks that SAM outperforms the Adam baseline by 4.76% on average and that strongest SAM variants outperform the Adam baseline by 8.01%. This suggests that SAM can be used to improve zero-shot OOD generalization, and that the strongest SAM variants can be used for an even further improvement.

2. To understand these performance gains, in Section 3.3 we provide a theoretical analysis of SAM under the distribution shift setting, with an OOD generalization bound based on sharpness in Lemma 1 derived in Section 3.3.

3. Next, in Section 4.2, we extend the setting considered above to gradual domain adaptation, where we have a sequence of unlabeled intermediate domains leading from the source to target domain. Our experiments across all four benchmarks demonstrate that SAM outperforms the Adam baseline by 0.82% on average and the strongest SAM variants achieve an even greater 1.52% average improvement over Adam, suggesting that SAM and the strongest SAM variants can be used for consistent performance gains in GDA as well.

4. Finally, in Section 4.4, we provide a bound extending Lemma 1 to the GDA setting and compare it to the prior bound in Wang et al. (2022) in Section 4.5. Although it is asymptotically the same as prior work in the GDA literature (Wang et al., 2022), we present several potential avenues for tightening this bound in future work in Section 4.6.

## 2 Preliminaries

**Classifier and Loss**  We consider a parametric model family $\Theta \subset \mathbb{R}^k$ with each model defined with respect to a specific choice of parameter $\theta \in \Theta$. The classifier induced from a model $\theta$ is denoted $f_\theta : \mathcal{X} \to \mathcal{Y}$. We consider bounded loss functions $\ell : \mathcal{Y} \times \mathcal{X} \times \Theta \to [0,1]$[1] and define the population and empirical risk with respect to (w.r.t.) parameters $\theta \in \Theta$ as $\mathcal{E}(\theta) := \mathbb{E}_{(\mathbf{x},y) \sim \mu}[\ell(y, \mathbf{x}, \theta)]$, for some $\mu \in \Delta(\mathcal{X} \times \mathcal{Y})$ and $\hat{\mathcal{E}}(\theta) := \frac{1}{n} \sum_{i=1}^{n} \ell(y_i, \mathbf{x}_i, \theta)$, respectively. We further define the sharpness of $\theta$ and the corresponding $\rho$-robust empirical loss following the standard definitions given in Foret et al. (2021).

**Definition 1** ($\rho$-Robust Risk). The $\rho$-robust risk of parameters $\theta \in \Theta$ is the maximum loss obtained by perturbing $\theta$ in the worst possible direction with $\ell_2$-norm[2] bounded by $\rho$:

$$\mathcal{E}^\rho(\theta) := \max_{\beta:\|\beta\|_2 \le \rho} \mathcal{E}(\theta + \beta) \tag{1}$$

Similarly, the empirical version of the $\rho$-robust risk is denoted $\hat{\mathcal{E}}^\rho(\theta)$.

**Definition 2** ($\rho$-Sharpness). The $\rho$-sharpness[3] of parameters $\theta \in \Theta$ measures how much the loss increases when we perturb them in the worst possible direction with $\ell_2$-norm bounded by $\rho$:

$$S^\rho(\theta) := \mathcal{E}^\rho(\theta) - \mathcal{E}(\theta)$$

**Sharpness-Aware Minimization**  Introduced by Foret et al. (2021), sharpness-aware minimization (SAM) proposes minimizing the $\rho$-robust empirical loss rather than the standard loss. Thus, SAM's objective is to find a minimizer $\theta^\star$ of the form:

$$\theta^\star = \operatorname{argmin}_{\theta \in \Theta} \max_{\beta:\|\beta\|_2 \le \rho} \mathcal{E}(\theta + \beta) \tag{2}$$

To solve the inner maximization $\operatorname{argmax}_{\beta:\|\beta\|_2 \le \rho} \mathcal{E}(\theta + \beta)$, two approximations are made:

---

[1]Our results can be easily extended to the case where the loss is bounded in range $[0, M]$ for $M > 0$ instead.

[2]Our results can be extended to any $\ell_p$-norm, but we have chosen $\ell_2$ since Foret et al. (2021) found $\ell_2$ sharpness to lead to the best results.

[3]This should not be confused with $m$-sharpness from Foret et al. (2021), where $m$ is the batch size used for training.

---

**Algorithm 1** Sharpness-Aware Minimization (SAM) with Variant-Specific Oracles

---

1: **Inputs:** Training set $S = \{(\mathbf{x}_i, y_i)\}_{i=1}^n$, Model parameter space $\Theta \subset \mathbb{R}^k$, Classifier $f_\theta : \mathcal{X} \to \mathcal{Y}$, Loss function $\ell : \mathcal{Y} \times \mathcal{X} \times \Theta \to [0,1]$, Learning rate $\eta > 0$, Perturbation radius $\rho > 0$, Batch size $b$, Variant-Specific gradient oracle $g$, Variant-Specific perturbation oracle $p$, Variant-Specific descent step oracle $a$

2: **Outputs:** Optimized model parameters $\theta_t$

3: Initialize $\theta_0$, $t \leftarrow 0$

4: **while** not converged **do**

5:     Sample mini-batch $\mathcal{B} = \{(x_1, y_1), \ldots, (x_b, y_b)\}$

6:     Compute the gradient $\nabla_\theta \hat{\mathcal{E}}(\theta)$ using the variant-specific gradient oracle:

$$\nabla_\theta \hat{\mathcal{E}}(\theta) = g(\mathcal{B}, \theta)$$

7:     Compute the perturbation $\beta^\star(\theta)$ using the variant-specific perturbation oracle:

$$\beta^\star(\theta) = p(\nabla_\theta \hat{\mathcal{E}}(\theta), \rho)$$

8:     Use the variant-specific descent step oracle to update the parameters according to

$$\theta_{t+1} \leftarrow \theta_t - \eta \cdot a(\mathcal{B}, \theta, \beta^\star(\theta_t))$$

9:     Increment $t \leftarrow t + 1$

10: **end while**

11: **return** $\theta_t$

---

1. A first-order Taylor expansion of the loss is used to approximate this max:

$$\underset{\beta : \|\beta\|_2 \leq \rho}{\arg\max} \mathcal{E}(\theta + \beta) \overset{(i)}{\approx} \underset{\beta : \|\beta\|_2 \leq \rho}{\arg\max} \mathcal{E}(\theta) + \beta^\top \nabla_\theta \mathcal{E}(\theta) \tag{3}$$

$$= \underset{\beta : \|\beta\|_2 \leq \rho}{\arg\max} \beta^\top \nabla_\theta \mathcal{E}(\theta) \tag{4}$$

$$= \frac{\rho \nabla_\theta \mathcal{E}(\theta)}{\|\nabla_\theta \mathcal{E}(\theta)\|} =: \beta^\star(\theta) \tag{5}$$

2. Then, the SAM gradient drops a second-order term arising from chain rule:

$$\nabla_\theta \mathcal{E}(\theta + \beta^\star(\theta)) = \frac{d(\theta + \beta^\star(\theta))}{d\theta} \nabla_\theta \mathcal{E}(\theta)|_{\theta + \beta^\star(\theta)} \tag{6}$$

$$= \nabla_\theta \mathcal{E}(\theta)|_{\theta + \beta^\star(\theta)} + \frac{d\beta^\star(\theta)}{d\theta} \nabla_\theta \mathcal{E}(\theta)|_{\theta + \beta^\star(\theta)} \tag{7}$$

$$\overset{(ii)}{\approx} \nabla_\theta \mathcal{E}(\theta)|_{\theta + \beta^\star(\theta)} \tag{8}$$

## 2.1 Background on SAM Variants

In Algorithm 1, we provide flexible algorithmic psuedocode for SAM with variant-specific oracles to account for the fact that the SAM variants compute the gradients, perturbations, and final descent steps in different ways. Below, we provide a brief explanation of the modifications each of the SAM variants makes to the gradients, perturbations, and final descent steps.

1. **SAM** (Foret et al., 2021): This is the original version of SAM, which uses the first-order approximation for the perturbation, as detailed below.

$$\beta^{\text{SAM}}(\theta) := \frac{\rho \nabla_\theta \mathcal{E}(\theta)}{\|\nabla_\theta \mathcal{E}(\theta)\|} \tag{9}$$

2. Adaptive SAM (**ASAM**) (Kwon et al., 2021): This is a version of SAM that uses a scale-invariant version of the first-order approximation:

$$\beta^{\text{ASAM}}(\theta) := \rho \frac{T_\theta^2 \nabla_\theta \mathcal{E}(\theta)}{\|T_\theta \nabla_\theta \mathcal{E}(\theta)\|} \tag{10}$$

The term $T_\theta$ refers to a normalization operator adaptive to each parameter's scale. Following the original paper and implementation provided by Samuel (2020), we set $T_\theta$ to be a diagonal matrix with $T_\theta^i = |\theta_i|$ for weight parameters and $T_\theta^i = 1$ for bias parameters for $i \in [k]$. Like SAM, Adaptive SAM has one hyperparameter $\rho$.

3. **FisherSAM** (Kim et al., 2022): FisherSAM computes the perturbation in the following way:

$$\beta^{\text{FisherSAM}}(\theta) := \rho \frac{f(\theta)^{-1} \nabla_\theta \mathcal{E}(\theta)}{\sqrt{\nabla_\theta \mathcal{E}(\theta) f(\theta)^{-1} \nabla_\theta \mathcal{E}(\theta)}} \tag{11}$$

This can be thought of as a special case of ASAM (Kwon et al., 2021) with the normalization operator $T_\theta$ set to be a diagonal matrix with entires $T_\theta^i = 1/\sqrt{1 + \eta f(\theta)_i}$, where, for $i \in [k]$, $f(\theta)_i := (\partial_{\theta_i} \hat{\mathcal{E}}(\theta))^2$ is the square of the batch gradient sum, a common and efficient approximation of the Fisher information matrix (Bottou et al., 2018; Khan et al., 2018). FisherSAM has two hyperparameters: the perturbation radius $\rho$ and the fisher coefficient $\eta$.

4. **K-SAM** (Ni et al., 2022): This is a variant of SAM that only uses the top $K$ data samples with the highest loss for both its gradient evaluations: both the perturbation gradient and the descent step gradient. K-SAM has two hyperparameters: the perturbation radius $\rho$ and the number of data points to use for gradient evaluations $K$.

5. **LookSAM** (Liu et al., 2022): This is a variant of SAM that only computes the gradient at the perturbed location every $L$ steps. For all time steps $T$ such that $T\%L = 0$ it performs the standard SAM update, i.e., $\theta' \leftarrow \theta - \eta \nabla_\theta \mathcal{E}(\theta)|_{\theta + \beta^\star}$, and computes the following:

$$g_v := \nabla_\theta \mathcal{E}(\theta)|_{\theta + \beta^\star} - \nabla_\theta \mathcal{E}(\theta) \left( \frac{\langle \nabla_\theta \mathcal{E}(\theta), \nabla_\theta \mathcal{E}(\theta)|_{\theta + \beta^\star} \rangle}{\|\nabla_\theta \mathcal{E}(\theta)\|^2} \right) \tag{12}$$

For all other time steps $T$ such that $T\%L \neq 0$, LookSAM calculates

$$g_s := \nabla_\theta \mathcal{E}(\theta) + \alpha \frac{\|\nabla_\theta \mathcal{E}(\theta)\|}{\|g_v\|} g_v \tag{13}$$

and uses $g_s$ for the descent step, i.e., $\theta' \leftarrow \theta - \eta g_s$. The idea behind Equation (12) is to remove the component of the descent gradient $\nabla_\theta \mathcal{E}(\theta)|_{\theta + \beta^\star}$ lying along the ascent gradient $\nabla_\theta \mathcal{E}(\theta)$, which is used to approximate $\beta^\star$ in Equation (9), so that for all time steps that reuse the previous descent gradient, the reused term $g_v$ disregards the component of the previous descent gradient lying along the previous ascent gradient. The term $\frac{\|\nabla_\theta \mathcal{E}(\theta)\|}{\|g_v\|}$ in Equation (13) is used to adaptively scale $\alpha$ so that $g_v$ is similar in magnitude to $\nabla_\theta \mathcal{E}(\theta)$.

6. **FriendlySAM** (Kim et al., 2022): This is a variant of SAM that uses a perturbation projected along the orthogonal complement of the full-batch gradient, following the observation that using only the full-batch direction of the ascent-step gradient impairs performance. The projection can be computed according to

$$\text{Proj}_{\nabla_\theta \mathcal{E}(\theta)}^\top \nabla_\theta \hat{\mathcal{E}}(\theta) = \nabla_\theta \hat{\mathcal{E}}(\theta) - s \nabla_\theta \mathcal{E}(\theta) \tag{14}$$

where $s = \cos(\nabla_\theta \mathcal{E}(\theta), \nabla_\theta \hat{\mathcal{E}}(\theta))$. However, since computing the full-batch gradient $\nabla_\theta \mathcal{E}(\theta)$ is computationally prohibitive, an exponentially moving average (EMA) is used in practice. Therefore, FriendlySAM uses the following perturbation:

$$\beta^{\text{FriendlySAM}}(\theta) := \rho \frac{\nabla_\theta \hat{\mathcal{E}}(\theta) - \sigma(\phi m + (1 - \phi) \nabla_\theta \hat{\mathcal{E}}(\theta))}{\|\nabla_\theta \hat{\mathcal{E}}(\theta) - \sigma(\phi m + (1 - \phi) \nabla_\theta \hat{\mathcal{E}}(\theta))\|} \tag{15}$$

Table 1: A comparison of the computational costs for each SAM variant. We assume each optimizer is run for $T$ epochs on a dataset with $N$ batches of batch size $B$. The computational cost is compared across 1) the total number of gradient evaluations throughout the trajectory, 2) the total number of data points used for gradient evaluations throughout the trajectory, and 3) the network layers the perturbation is applied to.

| Optimizer | Total Number Gradient Evals | Number Data Points Evaluated On | Network Layers Perturbation is Applied to |
|---|---|---|---|
| **SAM** | $2T$ | $B \times TN$ | All |
| **ASAM** | $2T$ | $B \times TN$ | All |
| **FisherSAM** | $2T$ | $B \times TN$ | All |
| **K-SAM** | $2T$ | $K \times TN$ | All |
| **LookSAM** | $T + T/L$ | $B \times TN$ | All |
| **FriendlySAM** | $2T$ | $B \times TN$ | All |
| **NoSAM** | $2T$ | $B \times TN$ | Only normalization layers |
| **ESAM** | $2T$ | $S \times TN, \text{for } S = f(\gamma) < B$ | Each layer w.p. $\xi$ |

where $m$ is the value of the EMA from the previous iteration. FriendlySAM has three hyperparameters: the perturbation radius $\rho$, the EMA momentum factor $\phi$, and the projection cosine similarity value $\sigma$.

7. **NoSAM** (Mueller et al., 2024): This is a variant of SAM that only performs the SAM perturbation on normalization layers.

8. EfficientSAM (**ESAM**) (Du et al., 2022): This is a variant of SAM intended to make SAM more efficient by applying two techniques to SAM. First, it performs *stochastic weight perturbation*, only performing the SAM perturbation on $\xi \in (0, 1)$ percent of the weights. Second, it performs *sharpness-sensitive data selection*, only computing gradients over data samples with the highest increase in loss after applying the perturbation. Thus, ESAM has three hyperparameters: the perturbation radius $\rho$, the sharpness sensitive data selection hyperparameter $\gamma$, and the stochastic weight parameter $\xi$.

In Table 1, we provide a comparison of the computational costs for each of the eight SAM variants defined above.

## 3 A Study of SAM for Zero-Shot OOD Generalization

In this section, we perform experiments comparing all eight of the previous SAM variants for zero-shot OOD generalization in Section 3.1, discuss the key takeaways from the experiments in Section 3.2, and perform a theoretical analysis of SAM for OOD generalization in Section 3.3.

**Zero-Shot OOD Generalization** In zero-shot OOD generalization, a model $\theta_S$ is trained on a source domain $S \in \Delta(\mathcal{X} \times \mathcal{Y})$ with a training set $\{(\mathbf{x}_i, y_i)\}_{i=1}^n$ drawn i.i.d. from $S$. Then, $\theta_S$ is evaluated on a test set drawn i.i.d. from a target domain $T \in \Delta(\mathcal{X} \times \mathcal{Y})$. The zero-shot OOD generalization error is given by $\mathcal{E}_T(\theta_S)$.

### 3.1 Experiments Comparing SAM Variants for Zero-Shot OOD Generalization

**Datasets** We use the same datasets as Wang et al. (2022), with a brief description of each given below.

a) **Color Shifted MNIST:** A dataset based on the original MNIST dataset (LeCun & Cortes, 1998). The source domain contains 50K images, whose pixels are normalized to be within $[0, 1]$. The pixel

Table 2: A comparison of zero-shot OOD accuracy for each of the SAM variants. In each column, the variant with the best performance is **bolded**. SAM outperforms the Adam baseline by 4.76% on average, while the strongest SAM variants outperform the baseline by 8.01% on average.

| Optimizer | Rotated MNIST | Color MNIST | Covertype | Portraits |
|---|---|---|---|---|
| **Adam** | $25.26_{\pm 1.79}$ | $84.97_{\pm 4.87}$ | $69.75_{\pm 2.30}$ | $86.43_{\pm 1.25}$ |
| **SAM** | $26.10_{\pm 0.90}$ | $93.88_{\pm 1.53}$ | $72.31_{\pm 1.06}$ | $87.77_{\pm 0.82}$ |
| **ASAM** | $26.09_{\pm 0.07}$ | $93.65_{\pm 1.18}$ | $72.70_{\pm 1.64}$ | $87.23_{\pm 0.81}$ |
| **FisherSAM** | $\mathbf{27.23}_{\pm 1.47}$ | $\mathbf{96.53}_{\pm 0.36}$ | $73.09_{\pm 3.33}$ | $88.67_{\pm 0.68}$ |
| **K-SAM** | $26.01_{\pm 1.98}$ | $95.36_{\pm 0.11}$ | $70.07_{\pm 2.04}$ | $87.73_{\pm 0.53}$ |
| **LookSAM** | $25.82_{\pm 2.00}$ | $96.34_{\pm 0.37}$ | $72.79_{\pm 2.06}$ | $88.07_{\pm 0.77}$ |
| **FriendlySAM** | $26.80_{\pm 0.73}$ | $95.61_{\pm 0.96}$ | $\mathbf{73.64}_{\pm 0.08}$ | $\mathbf{90.80}_{\pm 1.35}$ |
| **ESAM** | $25.80_{\pm 1.13}$ | $95.67_{\pm 1.30}$ | $72.37_{\pm 2.62}$ | $88.33_{\pm 1.30}$ |
| **NoSAM** | $24.81_{\pm 2.39}$ | $95.27_{\pm 0.56}$ | $72.53_{\pm 0.37}$ | $87.73_{\pm 0.68}$ |

values of each image are then increased by 1 to shift the range to $[1, 2]$ in the target domain. The intermediate domains are evenly distributed between source and target.

b) **Rotated MNIST:** A dataset of 50K images also based on LeCun & Cortes (1998). The source domain contains 50K images, each of which is rotated 60 degrees, to form the target domain. The intermediate domains are also evenly distributed between source and target.

c) **Cover Type:** This tabular dataset from Blackard & Dean (1999) contains 54 features used to predict the forest cover type at various locations. The data is sorted by distance to water body in ascending order, with the first 50K data points as the source domain, 10 intermediate domains with 40K data points each, and the target domain containing the final 50K data points.

d) **Portraits:** This dataset from Ginosar et al. (2017) contains photos of high school seniors from 1905 to 2013 with the goal of classifying the gender. The portraits are sorted chronologically, with the source domain as the first 2000 images, seven intermediate domains containing 2000 images each, and a target domain with the final 2000 images.

**Model Setup** For the computer vision tasks (Color Shifted MNIST, Rotated MNIST, and Portraits), we use a convolutional neural network (CNN) with 4 convolutional layers of 32 channels, followed by a fully-connected network (FCN) with 3 layers of 1024 neurons each and ReLU activation. For the tabular dataset (Cover Type), we use a multi-layer perceptron (MLP) with 3 layers of 256 neurons each. Models are trained using BatchNorm (Ioffe & Szegedy, 2015), DropOut(0.5) (Srivastava et al., 2014) and cross-entropy loss. When applying the SAM optimizer, we use Adam with the PyTorch default choices of learning rate of $10^{-2}$ and weight decay of $10^{-4}$ as a base optimizer. We also use a batch size of 128. When applying Adam, we use the PyTorch default choices of a learning rate of $10^{-3}$ and weight decay of $10^{-4}$, with a batch size of 128 as well. On Rotated MNIST, Color MNIST, and Portraits, we train for 100 epochs in the source and intermediate domains for both SAM and Adam. On Covertype, due to a larger dataset size and limited compute, we train for 25 epochs in the source and intermediate domains for all optimizers. Our full hyperparameter specification is given in Appendix B.

### 3.2 Discussion of Results

In Table 2, we report the zero-shot OOD accuracy on the $1 - \mathcal{E}_T(\theta_S)$ for each of the optimizers. Across all four datasets, the original SAM outperforms the Adam baseline by 4.76% on average, while the strongest SAM variants outperform the baseline by 8.01% on average. Variants such as LookSAM, F-SAM, and

FisherSAM offer the strongest and most consistent improvement over SAM. Among the variants with reduced computational cost (ESAM, LookSAM, K-SAM, and NoSAM), LookSAM and NoSAM perform the best, often outperforming original SAM in addition to being more efficient.

**Understanding the FisherSAM Performance Gains**   To recall, FisherSAM computes the perturbation according to:

$$\beta^{\text{FisherSAM}}(\theta) := \rho \frac{f(\theta)^{-1} \nabla_\theta \mathcal{E}(\theta)}{\sqrt{\nabla_\theta \mathcal{E}(\theta) f(\theta)^{-1} \nabla_\theta \mathcal{E}(\theta)}} \tag{16}$$

In contrast to the original SAM objective, FisherSAM takes the information geometry of the data into account, using an approximation $f(\theta)$ of the Fisher information matrix of parameters $\theta$ to find the maximum perturbation. Under the cross-entropy loss used specifically in the experiments in Table 2, the Fisher information matrix is directly equivalent to the Hessian matrix of the cross-entropy loss (Martens, 2020). Therefore, unlike SAM variants using first-order approximations, FisherSAM could be understood as a second-order approximation of the loss perturbation, hence leading to better generalization.

**Understanding the FriendlySAM Performance Gains**   Unlike FisherSAM, the connection between the FriendlySAM objective and OOD performance is not as explicit. To recall, FriendlySAM computes the perturbation according to

$$\beta^{\text{FriendlySAM}}(\theta) := \rho \frac{\nabla_\theta \hat{\mathcal{E}}(\theta) - \sigma(\phi m + (1 - \phi) \nabla_\theta \hat{\mathcal{E}}(\theta))}{\|\nabla_\theta \hat{\mathcal{E}}(\theta) - \sigma(\phi m + (1 - \phi) \nabla_\theta \hat{\mathcal{E}}(\theta))\|} \tag{17}$$

where $m$ is the value of the EMA from the previous iteration, $\phi$ is the EMA momentum factor, $\sigma$ is the cosine similarity value, and $\rho$ is the perturbation radius. In Li et al. (2024), the authors note that this modified perturbation makes FriendlySAM significantly more robust to the choice of the $\rho$ hyperparameter because it allows for penalizing the sharpness of the current mini-batch more directly, while disregarding the sharpness of the full-batch gradient. Since the optimal value of $\rho$ depends intricately on the choice of dataset and since we only tested $\rho \in \{0.01, 0.02, 0.05, 0.1, 0.2\}$, we conjecture that the performance gains from FriendlySAM in our experiments are due to penalizing sharpness in a more adaptive and stable way.

## 3.3   Theoretical Analysis of SAM for OOD Generalization

To start, we define the Wasserstein distance, which we use to capture the distance between probability distributions.

**Definition 3** (Wasserstein Distance)**.** The Wasserstein distance between two distributions $\mu$ and $\nu$ over $\mathcal{S} \subset \mathbb{R}^d$ is the smallest cost, as measured by some distance metric $d(\cdot, \cdot) : \mathcal{S} \times \mathcal{S} \to \mathbb{R}$, of moving mass between distributions $\mu$ and $\nu$, obtained by taking the infimum over the set of all joint distributions $\gamma \in \Gamma(\mu, \nu)$ with marginals $\mu$ and $\nu$:

$$W_p(\mu, \nu) := \inf_{\gamma \in \Gamma(\mu, \nu)} \left( \int_{\mathcal{S} \times \mathcal{S}} d(x, y)^p \mathrm{d}\gamma(x, y) \right)^{1/p} \tag{18}$$

Furthermore, we assume that the loss function is Lipschitz in its arguments, which holds under many commonly used loss functions, including the logistic loss, hinge loss, and squared loss when the input space is bounded.

**Assumption 1** (Lipschitz Loss)**.** For loss functions considered in this paper $\ell : \mathcal{Y} \times \mathcal{X} \times \Theta \to [0, 1]$, we assume there exist constants $\rho_1, \rho_2, \rho_3 > 0$ satisfying:

$$|\ell(y_1, \cdot, \cdot) - \ell(y_2, \cdot, \cdot)| \leq \rho_1 |y_1 - y_2|, \ \forall y_1, y_2 \in \mathcal{Y} \tag{19}$$
$$|\ell(\cdot, \mathbf{x_1}, \cdot) - \ell(\cdot, \mathbf{x_2}, \cdot)| \leq \rho_2 \|\mathbf{x_1} - \mathbf{x_2}\|, \ \forall \mathbf{x_1}, \mathbf{x_2} \in \mathcal{X}$$
$$|\ell(\cdot, \cdot, \theta_1) - \ell(\cdot, \cdot, \theta_2)| \leq \rho_3 \|\theta_1 - \theta_2\|, \ \forall \theta_1, \theta_2 \in \Theta$$

Now, we provide a lemma bounding the error difference over shifted domains for SAM, which is a "sharpness-aware" version of Lemma 1 in Wang et al. (2022). Our result bounds the absolute difference between the $\rho$-robust error of a model $\theta_\mu$ in domain $\mu$ and the error of a model $\theta_\nu$ in domain $\nu$ as function of the sharpness of the model in domain $\mu$, the distance between the models, and the Wasserstein distance between the domains $\mu$ and $\nu$. This result holds for any choice of $\mu, \nu$ on $\mathcal{Y} \times \mathcal{X}$.

**Lemma 1** (Sharpness-Aware Error Difference Over Shifted Domains). Given an error function $\mathcal{E}_\mu(\theta) := \mathbb{E}_{(\mathbf{x}, y) \sim \mu}[\ell(y, \mathbf{x}, \theta)]$ with loss satisfying Assumption 1 and any distributions $\mu, \nu$ on $\mathcal{Y} \times \mathcal{X}$, we have that

$$|\mathcal{E}_\mu^\rho(\theta_\mu) - \mathcal{E}_\nu(\theta_\nu)| \leq S^\rho(\theta_\mu) + \mathcal{O}(\|\theta_\mu - \theta_\nu\| + W_p(\mu, \nu)) \tag{20}$$

Before stating our main result of this section in Theorem 1, we state the PAC-Bayesian generalization bound introduced in Foret et al. (2021) which bounds the population error of a model $\theta$ in terms of its empirical sharpness.

**Lemma 2** (Sharpness-Aware Generalization Bound). For any model $\theta \in \Theta \subset \mathbb{R}^k$ satisfying $\mathcal{E}(\theta) \leq \mathbb{E}_{\epsilon \sim \mathcal{N}(0, \rho^2 I)}[\mathcal{E}(\theta + \epsilon)]$ for some $\rho > 0$, then w.p. $\geq 1 - \delta$,

$$\mathcal{E}(\theta) \leq \hat{\mathcal{E}}^\rho(\theta) + \mathcal{O}\left(\sqrt{\frac{k \ln(\|\theta\|_2^2 / \rho^2) + \ln(n/\delta)}{n}}\right) \tag{21}$$

Finally, using the error difference lemma from Lemma 1 and the PAC-Bayesian bound from Lemma 2, we are able to state an OOD generalization bound for SAM for any choice of distributions $\mu, \nu$ on $\mathcal{Y} \times \mathcal{X}$. This result upper bounds the error of the empirically obtained SAM solution $\hat{\theta}_\mu$ on domain $\mu$ by the error of the population robust risk minimizer $\theta_\nu$ over domain $\nu$, the sample complexity term from the PAC Bayesian analysis in Lemma 2, the sharpness of $\theta_\mu$ in domain $\mu$, the distance between $\theta_\mu$ and $\theta_\nu$, and the Wasserstein distance between distributions $\mu$ and $\nu$.

**Theorem 1** (Sharpness-Aware Domain Adaptation Error). Given distributions $\mu, \nu$ over $\mathcal{X} \times \mathcal{Y}$, an error function $\mathcal{E}$ with loss satisfying Assumption 1, the empirical SAM solution $\hat{\theta}_\mu$ in domain $\mu$ satisfying $\mathcal{E}_\mu(\hat{\theta}_\mu) \leq \mathbb{E}_{\epsilon \sim \mathcal{N}(0, \rho^2 I)}[\mathcal{E}_\mu(\hat{\theta}_\mu + \epsilon)]$ for some $\rho > 0$ [4], the population robust risk minimizer $\theta_\nu = \arg\min_{\theta \in \Theta} \mathcal{E}_\nu^\rho(\theta)$ in domain $\nu$, and the population robust risk minimizer $\theta_\mu = \arg\min_{\theta \in \Theta} \mathcal{E}_\mu^\rho(\theta)$ in domain $\mu$, then w.p. $\geq 1 - \delta$,

$$\mathcal{E}_\mu(\hat{\theta}_\mu) \leq \mathcal{E}_\nu(\theta_\nu) + \mathcal{O}\left(\frac{\sqrt{k \ln(\|\theta_\mu\|_2^2 / \rho^2) + \ln(n/\delta)}}{\sqrt{n}} + \|\theta_\mu - \theta_\nu\| + W_p(\mu, \nu)\right) + S^\rho(\theta_\mu)$$

## 4 A Study of SAM for Gradual Domain Adaptation

In this section, we study SAM applied to a related OOD generalization setting called gradual domain adaptation (GDA), where self-training is performed on intermediate domains between the source and target domain to improve the target domain accuracy (Kumar et al., 2020; Wang et al., 2022; Zhuang et al., 2024; He et al., 2024). We provide an introduction to GDA in Section 4.1, describe our experimental setup in Section 4.2, discuss our experimental results in Section 4.3, provide a generalization bound based on sharpness for GDA in Section 4.4, compare our bound to that of the prior work Wang et al. (2022) in Section 4.5, and discuss several potential avenues for obtaining a tighter bound in Section 4.6.

### 4.1 Gradual Domain Adaptation (GDA)

**Gradually Shifting Distributions** We adopt the settings of Wang et al. (2022) and Kumar et al. (2020), where we have $T + 1$ gradually shifting distributions indexed by $\{0, 1, ..., T\}$, with 0 corresponding to the source domain. Each domain $t \in [T]$ is a distribution $\mu_t$ over $\mathcal{X} \times \mathcal{Y}$ and we start with a training distribution $\{(\mathbf{x}_i, y_i)\}_{i=1}^{n_0}$ drawn i.i.d. from the source domain $\mu_0$. To ease the presentation, we assume that each subsequent domain has $n$ unlabeled training examples. We measure distribution shifts according to the Wasserstein distance and define the following shorthand notation.

---

[4]This assumption means that adding Gaussian perturbation around $\hat{\theta}_\mu$ increases the expected loss, which should hold for local minima (Foret et al., 2021).

---

**Algorithm 2** Gradual Self Training (GST) with SAM

---

1: **Inputs:** Labeled source data $S_0 := \{(x_i, y_i)\}_{i=1}^{n_0}$ and unlabeled data $(S_t)_{t=1}^T := (\{x_i^{(t)}\}_{i=1}^n)_{t=1}^T$ from intermediate domains; perturbation radius $\rho$; weight decay parameter $\lambda$.

2: Train model on labeled data from the source domain according to the SAM objective:

$$\theta_0 := \mathrm{argmin}_{\theta' \in \Theta} \max_{\|\beta\| \leq \rho} \sum_{i=1}^{n_0} \ell(y_i, x_i, \theta' + \beta) + \lambda\|\theta'\|$$

3: **for** $t = 1$ **to** $T$ **do**

4:     Generate psuedo-labels $\hat{y}_i := f_{\theta_{t-1}}(x_i^{(t)})$.

5:     Self-train the model using psuedo-labels according to the SAM objective:

$$\theta_t = \mathrm{argmin}_{\theta' \in \Theta} \max_{\|\beta\| \leq \rho} \sum_{i=1}^n \ell(\hat{y}_i, x_i^{(t)}, \theta' + \beta) + \lambda\|\theta'\|$$

6: **end for**

7: **return** $\theta_T$

---

**Definition 4** (Distribution Shifts)**.** The distribution shift between successive pair of gradually shifted distributions and the average distribution shift of all pairs are, respectively, defined as $\forall t \in [T-1]$:

$$\Delta_t := W_p(\mu_{t+1}, \mu_t), \quad \text{and} \quad \Delta := \frac{1}{T} \sum_{t=0}^{T-1} \Delta_t \tag{22}$$

**Gradual Domain Adaptation**   In gradual domain adaptation (GDA), a learner is given $n_0$ labeled examples from a source domain with index $t = 0$, and then given sequential access to $n$ unlabeled examples from domains $t = \{1, 2, ..., T\}$:

$$S_t := \{(x_i^{(t)})\}_{i=1}^n, \quad t \in \{1, 2, \ldots, T\}$$

The goal is to gradually train a classifier on the intermediate domains using psuedo-labels to minimize the generalization error in the target domain $\mathcal{E}_T(\theta_T)$, as specified in Algorithm 2.

### 4.2   Experiments Comparing SAM Variants for GDA

In this section, we perform an empirical study of SAM for GDA using the same four datasets as in Section 3.

**Datasets and Model Setup**   We use the same datasets and model setup as our earlier experiment, as detailed in Section 3.1. As in the experiments from Section 3, on Rotated MNIST, Color MNIST, and Portraits, we train for 100 epochs in the source domain for all optimizers, and on Covertype, we train for 25 epochs in the source domain. The results of this experiment are presented in Figure 1.

**SAM Variant Setup**   We use the same hyperparameter setup for the SAM variants as given in Appendix B. Due to limited compute, we do not perform the entire ablation of intermediate domains for all of the SAM variants. Instead, we choose the optimal number of intermediate domains $T^\star$ for SAM from Figure 1 and obtain the results for this choice of $T^\star$.

### 4.3   Discussion of Results

In Figure 1, we report the target domain accuracy $1 - \mathcal{E}_T(\theta_T)$ for each of the optimizers. Our results reveal that SAM outperforms Adam (SAM with $\rho = 0$) on all datasets: by 1.03% on Covertype, by 0.96% on Portraits, by 0.75% on Rotated MNIST, and by 0.53% on Color MNIST. This suggests that SAM can be applied to GDA to consistently achieve stronger performance. For most of the experimental datasets, any

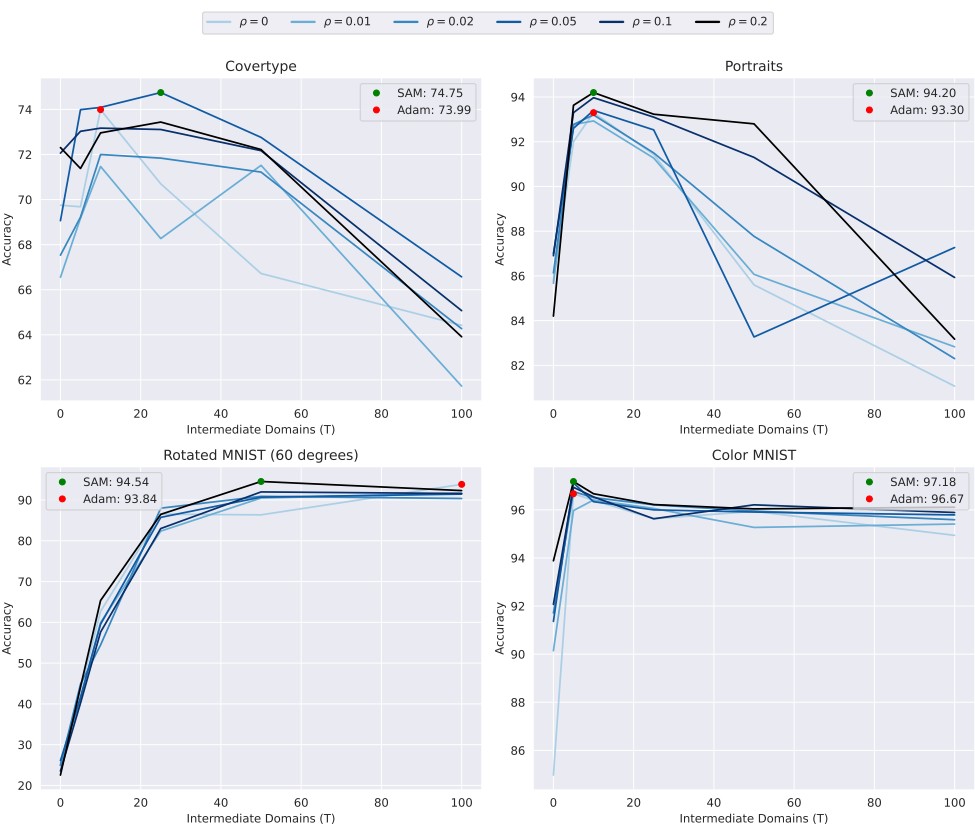

Figure 1: A comparison of SAM with varying levels of perturbation strength $\rho \in \{0, 0.01, 0.02, 0.05, 0.1, 0.2\}$, with $\rho = 0$ corresponding to Adam, on GDA datasets with varying levels of intermediate domains $T$. **SAM outperforms Adam on all datasets**: by 1.03% on Covertype, by 0.96% on Portraits, by 0.75% on Rotated MNIST, and by 0.53% on Color MNIST.

value of $\rho \in \{0.01, 0.02, 0.05, 0.1, 0.2\}$ tends to outperform Adam, suggesting that SAM is not too sensitive to the perturbation radius hyperparameter. On Rotated MNIST, Portraits, and Color MNIST, SAM with $\rho = 0.2$ leads to the strongest performance and the improvement is relatively monotonic in $\rho$, while on Covertype, SAM with $\rho = 0.05$ leads to the strongest performance.

In our study of SAM variants for GDA in Table 3, we find that the strongest SAM variants lead to an average improvement of 1.42% compared to using Adam, in comparison to SAM, which only offers a 0.82% improvement. In contrast to the zero-shot OOD generalization results from Section 3 in Table 2 where the SAM variants with reduced computational cost generally outperformed Adam, the SAM variants with reduced computational cost (K-SAM, LookSAM, NoSAM, and ESAM) tend to underperform Adam on GDA. Once again, FriendlySAM performs well, outperforming SAM by 0.40% on average across all datasets; however, FisherSAM slightly underperforms SAM by 0.01% this time.

In general, these results suggest that SAM can be used to consistently improve target domain error for GDA, and the strongest variant Friendly SAM can lead to even further improvements compared to SAM.

## 4.4 Generalization Bound for SAM for GDA

In this section, we provide an extension of Theorem 1 to the related setting of gradual domain adaptation (GDA), a technique which improves target domain error by performing gradual self-training (GST) on unlabeled intermediate domains between the source and target domain, for which the average distribution

Table 3: A comparison of GDA accuracy for each of the SAM variants. For reference, the top accuracy obtained by Adam and SAM is also provided. In each column, the variant with the best performance is **bolded**.

| Optimizer | Rotated MNIST (T=50) | Color MNIST (T=10) | Covertype (T=25) | Portraits (T=10) |
|---|---|---|---|---|
| **Adam** | $93.84_{\pm 1.64}$ | $96.67_{\pm 1.87}$ | $73.99_{\pm 0.30}$ | $93.30_{\pm 1.17}$ |
| **SAM** | $94.54_{\pm 0.60}$ | $97.18_{\pm 1.53}$ | $74.75_{\pm 0.06}$ | $94.20_{\pm 0.44}$ |
| **ASAM** | $93.69_{\pm 0.51}$ | $97.15_{\pm 0.24}$ | $74.40_{\pm 1.35}$ | $93.73_{\pm 0.66}$ |
| **FisherSAM** | $93.69_{\pm 2.37}$ | $97.40_{\pm 0.07}$ | $75.16_{\pm 1.14}$ | $94.40_{\pm 0.75}$ |
| **K-SAM** | $85.12_{\pm 5.72}$ | $97.76_{\pm 0.13}$ | $72.29_{\pm 1.51}$ | $93.23_{\pm 0.13}$ |
| **LookSAM** | $89.57_{\pm 2.28}$ | $\mathbf{97.79}_{\pm 0.13}$ | $73.15_{\pm 2.67}$ | $94.23_{\pm 0.34}$ |
| **FriendlySAM** | $\mathbf{94.81}_{\pm 0.05}$ | $97.35_{\pm 0.07}$ | $\mathbf{75.64}_{\pm 1.09}$ | $\mathbf{94.47}_{\pm 0.17}$ |
| **ESAM** | $93.23_{\pm 0.24}$ | $97.38_{\pm 0.40}$ | $72.39_{\pm 2.64}$ | $94.23_{\pm 0.39}$ |
| **NoSAM** | $94.60_{\pm 0.83}$ | $97.06_{\pm 0.07}$ | $72.85_{\pm 1.61}$ | $94.27_{\pm 0.19}$ |

shift is small (Kumar et al., 2020; Wang et al., 2022). The algorithm that performs GDA by GST using SAM is presented in Algorithm 2.

**GDA as Online Learning** For completeness, we briefly describe the online learning framework from Wang et al. (2022) we will use to obtain this bound.

**Definition 5** (Discrepancy Measure). Given a model family $\Theta$, input space $\mathcal{X}$, output space $\mathcal{Y}$, corresponding loss function $\ell : \mathcal{Y} \times \mathcal{X} \times \Theta \to [0, 1]$, and a $(t + 1)$-dimensional probability vector $\mathbf{q}_t := (q_0, q_1, \ldots, q_t)$, the discrepancy measure is given by

$$\mathrm{disc}(\mathbf{q}_t) := \sup_{\theta \in \Theta} \left( \mathcal{E}_t(\theta) - \sum_{k=0}^{t-1} q_k \mathcal{E}_k(\theta) \right) \tag{23}$$

Intuitively, the discrepancy measure captures the non-stationarity of the gradually shifting data, as measured by the expressivity of the model class and corresponding loss function (Kuznetsov & Mohri, 2020a). Next, we define the sequential Rademacher complexity, which generalizes the standard Rademacher complexity to the online learning setting (Rakhlin et al., 2015).

**Definition 6** (Complete Binary Trees and Tree Path). We define complete binary trees $\mathscr{X}$ and $\mathscr{Y}$ of depth $T + 1$ and a corresponding path in either $\mathscr{X}$ or $\mathscr{Y}$, respectively, as:

$$\begin{aligned} \mathscr{X} &:= (\mathscr{X}_0, \ldots, \mathscr{X}_T), \text{ with } \mathscr{X}_t : \{\pm 1\}^t \to \mathscr{X} , t \in [T] \\ \mathscr{Y} &:= (\mathscr{Y}_0, \ldots, \mathscr{Y}_T), \text{ with } \mathscr{Y}_t : \{\pm 1\}^t \to \mathscr{Y} , t \in [T] \\ \sigma &:= (\sigma_0, \ldots, \sigma_T) \in \{\pm 1\}^t \text{ in either } \mathscr{X} \text{ or } \mathscr{Y} \end{aligned} \tag{24}$$

**Definition 7** (Sequential Rademacher Complexity). Given a model family $\Theta$, input space $\mathcal{X}$, output space $\mathcal{Y}$, corresponding loss function $\ell : \mathcal{Y} \times \mathcal{X} \times \Theta \to [0, 1]$, $(t+1)$-dimensional probability vector $\mathbf{q}_t := (q_0, q_1, \ldots, q_t)$, and a binary tree path $\sigma$, we define the sequential Rademacher complexity as:

$$R_t^{\mathrm{seq}}(\Theta) := \sup_{\mathscr{X}, \mathscr{Y}} \mathbb{E}_\sigma \left[ \sup_{\theta \in \Theta} \sum_{k=0}^{t-1} \sigma_k q_k \ell(f_\theta(\mathscr{X}_k(\sigma)), \mathscr{Y}(\sigma)) \right]$$

**Reduction View of GDA** Moreover, we adopt the same reduction approach in Wang et al. (2022), where each of the $nT$ samples is viewed as the smallest element of the adaptation process, enabling generalization

bounds with an effective sample size of $nT$ rather than $T$. For further explanation on this view, see Wang et al. (2022). With this reduction, online-learning view, we can now state our generalization bound for GDA performed using SAM.

**Theorem 2** (Total Sharpness-Aware Error Under GDA). For any $\delta \in (0, 1)$, w.p. $\geq 1 - \delta$, the population risk of the gradually adapted model $\theta_T$ constructed from intermediate models $\theta_0, \ldots, \theta_{T-1}$ according to Algorithm 2, and satisfying $\forall t \in [T]$, $\mathcal{E}(\theta) \leq \mathbb{E}_{\epsilon \sim \mathcal{N}(0, \rho_t^2 I)}[\mathcal{E}(\theta_t + \epsilon)]$ for some $\rho_t > 0$[5], can be bounded according to:

$$\mathcal{E}_T(\theta_T) \leq \mathcal{E}_0(\theta_0) + \mathcal{O}\left(T(\Delta + \theta_{\text{avg}} + S_{\text{avg}}) + \frac{T(W_{\text{avg}} + \sqrt{\ln(n/\delta)})}{\sqrt{n}} + \sqrt{\frac{\ln(1/\delta)}{nT}} + \sqrt{\ln(nT)} \, \mathcal{R}_{nT}^{\text{seq}}(\Theta)\right)$$

where $\theta_{\text{avg}} := \frac{1}{T}\sum_{t=1}^{T} \|\theta_t - \theta_{t-1}\|_2$ is the average weight shift, $S_{\text{avg}} := \frac{1}{T}\sum_{t=1}^{T} S^{\rho_t}(\theta_t)$ is the average sharpness, and $W_{\text{avg}} := \frac{1}{T}\sum_{t=1}^{T} \mathcal{O}\left(\sqrt{k \ln(\|\theta_t\|_2^2/\rho^2)}\right)$ is the average weight norm.

**Proof Sketch** Our proof of Theorem 2 proceeds similarly to Wang et al. (2022). By applying Corollary 2 of Kuznetsov & Mohri (2020a), a preliminary bound on the error in the target domain $\mathcal{E}_T(\theta_T)$ can be obtained:

$$\mathcal{E}_T(\theta_T) \leq \sum_{t=0}^{T} \mathcal{E}_t(\theta_T) + \text{disc}(\mathbf{q}_{n(T+1)}) + \|\mathbf{q}_{n(T+1)}\|_2 \cdot \mathcal{O}\left(1 + \sqrt{\ln\frac{1}{\delta}}\right) + \mathcal{O}(\sqrt{\ln(nT)} \, \mathcal{R}_{nT}^{\text{seq}}(\Theta)) \tag{25}$$

Then, term one, $\sum_{t=0}^{T} \mathcal{E}_t(\theta_T)$, can be bounded by applying Theorem 1 successively for all $t \in [T-1]$. We defer discussion of the bounds on terms two and three and the full algebraic manipulations to Appendix A.

### 4.5 Comparison of Sharpness-Aware Bound With Standard Bound

The bound we obtain in Theorem 2 is of the form

$$\mathcal{E}_0(\theta_0) + \mathcal{O}\left(T(\Delta + \theta_{\text{avg}} + S_{\text{avg}}) + \frac{T(W_{\text{avg}} + \sqrt{\ln(n/\delta)})}{\sqrt{n}} + \frac{1}{\sqrt{nT}} + \sqrt{\frac{\ln(1/\delta)}{nT}} + \sqrt{\ln(nT)} \, \mathcal{R}_{nT}^{\text{seq}}(\Theta)\right)$$

whereas in Wang et al. (2022), the generalization upper bound obtained is of the form

$$\mathcal{E}_0(\theta_0) + \mathcal{O}\left(T\Delta + T\sqrt{\frac{\ln 1/\delta}{n}} + \frac{1}{\sqrt{nT}} + \sqrt{\frac{\ln(1/\delta)}{nT}} + \sqrt{\ln(nT)} \, \mathcal{R}_{nT}^{\text{seq}}(\Theta)\right) \tag{26}$$

Our analysis uses the original PAC Bayes generalization bound from Foret et al. (2021) in Lemma 2, which specifies a prior and posterior over the parameter space to bound the error directly as a function of the parameters. This is in contrast to the generalization bound applied in Wang et al. (2022), which uses only the Rademacher complexity of the predictor.

Since our analysis applies Lemma 2 successively to bound the sharpness-aware error difference over shifted domains with different optimal parameters, Theorem 1 must relate the parameters of the successive domains separately, which introduces the weight shift $\theta_{\text{avg}}$. In addition to capturing the average sharpness $S_{\text{avg}}$ term, the PAC Bayes bound introduces an additional weight norm term $W_{\text{avg}}$. Moreover, the PAC Bayesian bound yields a $\mathcal{O}\left(T\sqrt{\ln(n/\delta)}/\sqrt{n}\right)$ sample complexity term, versus the $\mathcal{O}\left(T\sqrt{\ln(1/\delta)}/\sqrt{n}\right)$ original analysis using the Rademacher complexity from Wang et al. (2022). While this discrepancy gets absorbed in the final asymptotic rate in terms of $n$ and $T$ (both ours and Wang et al. (2022) are $\mathcal{E}_0(\theta_0) + \tilde{\mathcal{O}}(T/\sqrt{n} + 1/\sqrt{nT})$), our analysis is looser non-asymptotically due to the three constants and the slightly worse poly-logarithmic sample complexity term. In light of this, we pose a conjecture for how this analysis can be tightened in future work.

---

[5]This assumption means that adding Gaussian perturbation to each solution $\theta_t$ increases the expected loss, which should hold for all $\theta_t$ obtained from executing SAM in Algorithm 2 (Foret et al., 2021)

### 4.6 Obtaining a Tighter Sharpness-Aware Error Analysis

In the proof of Theorem 2 in Appendix A, the only term that depends on the error difference between shifted domains is $\frac{1}{T+1} \sum_{t=0}^{T} \mathcal{E}_t(\theta_T)$, which is bounded by repeatedly applying Lemma 1 to domain pairs $(\mu_t, \mu_{t+1})$ for $t \in \{0, 1, \ldots, T-1\}$. Thus, we can expect the overall sample complexity to remain the same between our Theorem 2 and the main result in Wang et al. (2022). Assuming one continues to use the discrepancy based framework and the Sequential Rademacher Complexity from Kuznetsov & Mohri (2020b); Rakhlin et al. (2015), the only way to improve the analysis would be to provide a tighter error difference between shifted domains, as in Lemma 1, where we obtained

$$|\mathcal{E}_\mu^\rho(\theta_\mu) - \mathcal{E}_\nu(\theta_\nu)| \leq S^\rho(\theta_\mu) + \mathcal{O}(\|\theta_\mu - \theta_\nu\| + W_p(\mu, \nu))$$

We conjecture that one may be able to get a tighter bound for SAM through a *localized* analysis that analyzes the error difference between shifted domains specifically for the sharpness-aware minimization classifier, in contrast to the current *uniform* analysis which applies to every classifier in the model class $\Theta$. A localized analysis could exploit implicit properties of SAM, such as denoised features (Chen et al., 2024), lower-rank features (Andriushchenko et al., 2023), or balanced feature learning (Springer et al., 2024), to get a tighter bound. However, we leave development of a localized analysis framework for future work.

## 5   Related Works

**Sharpness and Generalization**   The study of the relationship between sharpness and generalization dates back to at least Hochreiter & Schmidhuber (1997), which motivates flat minima by a minimum description length argument. Since then, Keskar et al. (2017) have explored how various hyperparameter choices affect sharpness and Jiang et al. (2020) have found that sharpness is among the empirical measures most strongly correlated with generalization. However, Dinh et al. (2017) demonstrated that sharp minima can indeed generalize under reparameterizations which cause flat minima to become arbitrarily sharp. A follow-up work of this proposes a measure of sharpness tied to the information geometry of the data that is invariant under reparameterizations (Liang et al., 2019). Many other works have continued to explore algorithms that lead to flatter solutions (Foret et al., 2021; Wortsman et al., 2022; Chaudhari et al., 2019) – notably, SAM (Foret et al., 2021) – and empirical settings in which these algorithms work (Kaddour et al., 2022).

**Sharpness and OOD Generalization**   In the context of domain generalization (DG), Cha et al. (2021) propose a modified version of stochastic weight averaging (Izmailov et al., 2018) which leads to flatter minima with improved DG. They also provide generalization bounds that depend on the empirical robust loss in the source domain. Zhang et al. (2023) later introduce a flatness-aware minimization algorithm for DG which leads to improved performance, with theoretical results showing that their algorithm controls the maximum eigenvalue of the Hessian and indeed leads to flatter minima. Zou et al. (2024) present out-of-distribution (OOD) generalization bounds based on sharpness by using algorithmic robustness, refining the bounds presented in Cha et al. (2021). Finally, in a study of SAM's role in improving generalization for large language models, Bahri et al. (2022) has found that SAM can improve cross-lingual transfer and multi-task generalization, with significant gains in the limited training data regime.

**Sharpness-Aware Minimization**   Sharpness-Aware Minimization (SAM) was originally proposed in Foret et al. (2021), motivated by a PAC Bayesian analysis giving a generalization bound in terms of the expected sharpness over an isotropic Gaussian perturbation, i.e., $\mathbb{E}_{\beta \sim \mathcal{N}(0, \rho^2 I)}[\mathcal{E}(\theta + \beta)]$. The authors then upper bound this by the maximum and estimate the maximum using a first-order Taylor approximation: $\mathbb{E}_{\beta \sim \mathcal{N}(0, \rho^2 I)}[\mathcal{E}(\theta + \beta)] < \max_{\|\beta\| \leq \rho}[\mathcal{E}(\theta + \beta)] \approx \rho \frac{\nabla_\theta \mathcal{E}(\theta)}{\|\nabla_\theta \mathcal{E}(\theta)\|}$. The practical implementation of SAM uses this first-order approximation. Despite the original bound in terms of sharpness, Wen et al. (2023) have recently demonstrated that the flatness of the final solution does not sufficiently capture the generalization benefit from SAM alone, suggesting a more thorough study into implicit biases of SAM. Notably, SAM has been found to lead to lower rank features with fewer active ReLU units (Andriushchenko et al., 2023), to enhance feature quality by selecting more balanced features (Springer et al., 2024), to enhance robustness to label noise through implicitly regularizing the model Jacobian (Baek et al., 2024), and to have an implicit denoising mechanism which prevents harmful overfitting in settings when SGD would harmfully overfit (Chen

et al., 2024). Finally, many variants of SAM have been proposed to improve the efficiency and accuracy of the original SAM (Kwon et al., 2021; Kim et al., 2022; Mueller et al., 2024; Liu et al., 2022; Ni et al., 2022; Du et al., 2022; Li et al., 2024). In Section 2.1, we provide a detailed explanation of each, including a comparison of their computational costs in Table 1.

**Gradual Self-Training for GDA**  Kumar et al. (2020) first introduce gradual self-training (GST) in GDA, which outperforms standard self-training without intermediate domains. They also provide the first generalization bounds for GST in GDA; however, their bounds have an exponential dependence on the number of intermediate domains $T$, only hold for the ramp loss, and only hold for the Wasserstein distance of order $\infty$. Following this, Wang et al. (2022) remove this exponential dependence on $T$ with generalization bounds that depend on $T$ only linearly and additively. Wang et al. (2022) also generalize the analysis to any $\rho$-Lipschitz losses and Wasserstein distances of any order $p \geq 1$. These refined bounds also suggest the existence of an optimal choice of intermediate domains $T$, which Wang et al. (2022) derive as well. Subsequently, He et al. (2024) propose a new method of generating intermediate domains in an encoded feature space that are closer to the Wasserstein geodesic, leading to improved performance. Zhuang et al. (2024) later provide a continuous-time extension of He et al. (2024) using Wasserstein gradient flow.

# 6  Limitations and Future Work

The main limitation of this work is the discrepancy between our theoretical analysis based off sharpness, which is the same asymptotic rate as the prior work Wang et al. (2022), and our empirical results demonstrating consistent performance gains for SAM. At the end of Section 4.6, we posed a conjecture for how the analysis for SAM can be tightened. Future work can perform a more detailed theoretical analysis of SAM in order to explain the empirical benefits of using SAM for OOD generalization. Additionally, future work can attempt to theoretically explain the strong performance benefits of using SAM variants like FisherSAM and FriendlySAM for OOD generalization.

# 7  Conclusion

In this paper, we performed a theoretical and empirical study of SAM for OOD generalization. First, we experimentally compared eight SAM variants on zero-shot OOD generalization, finding that, across our four benchmarks, the original SAM achieved a 4.76% average improvement over the Adam baseline, while the strongest SAM variants achieved a 8.01% average improvement over the Adam baseline. Next, we derived an OOD generalization bound based on sharpness. Then, we experimentally compared the eight SAM variants on gradual domain adaptation (GDA), where intermediate domains are constructed between the source and target domain and iterative self-training is done on these intermediate domains to improve the target domain error. Our experiments found that, across our four benchmarks, the original SAM achieved a 0.82% average improvement over the Adam baseline, while the strongest SAM variants achieve a 1.42% average improvement over the Adam baseline. We provided an extension of our OOD generalization bound to get a generalization bound based on sharpness for GDA, which had the same asymptotic rate as the prior bound in Wang et al. (2022). This discrepancy between the theoretical and empirical results sheds light on the broader issue of giving tighter generalization bounds for SAM, especially in the OOD setting, to reconcile its consistent performance gains in practice. Our theoretical results provide a starting point for doing this, and our empirical results suggest that SAM can be used empirically to achieve significant gains for OOD generalization.

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

# A   Full Proofs

## A.1   Sharpness-Aware Error Difference Over Shifted Domains

**Lemma 1** (Sharpness-Aware Error Difference Over Shifted Domains)**.** Given an error function $\mathcal{E}_\mu(\theta) := \mathbb{E}_{(\mathbf{x},y)\sim\mu}[\ell(y,\mathbf{x},\theta)]$ with loss satisfying Assumption 1 and any distributions $\mu,\nu$ on $\mathcal{Y}\times\mathcal{X}$, we have that

$$|\mathcal{E}_\mu^\rho(\theta_\mu) - \mathcal{E}_\nu(\theta_\nu)| \leq S^\rho(\theta_\mu) + \mathcal{O}(\|\theta_\mu - \theta_\nu\| + W_p(\mu,\nu)) \tag{20}$$

*Proof.* Letting $\gamma(\mu,\nu)$ be any coupling of $\mu\times\nu$ and defining $\rho := \max\{\rho_1,\rho_2,\rho_3\}$ with $\rho_1,\rho_2,\rho_3$ in Assumption 1, we have that

$$|\mathcal{E}_\mu^\rho(\theta_\mu) - \mathcal{E}_\nu(\theta_\nu)| = \left|\mathcal{E}_\mu^\rho(\theta_\mu) - \mathcal{E}_\mu(\theta_\mu) + \mathcal{E}_\mu(\theta_\mu) - \mathcal{E}_\nu(\theta_\nu)\right| \tag{27}$$

$$\leq S^\rho(\theta_\mu) + |\mathcal{E}_\mu(\theta_\mu) - \mathcal{E}_\nu(\theta_\nu)| \tag{28}$$

$$= S^\rho(\theta_\mu) + |\mathbb{E}_{(x,y)\sim\mu}[\ell(y,x,\theta_\mu)] - \mathbb{E}_{(x,y)\sim\nu}[\ell(y,x,\theta_\nu)]| \tag{29}$$

$$= S^\rho(\theta_\mu) + \left|\int \ell(y,x,\theta_\mu)\mathrm{d}\mu - \int \ell(y',x',\theta_\nu)\mathrm{d}\nu\right| \tag{30}$$

$$\leq S^\rho(\theta_\mu) + \int |\ell(y,x,\theta_\mu) - \ell(y',x',\theta_\nu)|\,\mathrm{d}\gamma(\mu,\nu) \tag{31}$$

$$\leq S^\rho(\theta_\mu) + \rho\left(\int (|y-y'| + \|x-x'\| + \|\theta_\mu - \theta_\nu\|)\,\mathrm{d}\gamma(\mu,\nu)\right) \tag{32}$$

$$= S^\rho(\theta_\mu) + \rho\left(\|\theta_\mu - \theta_\nu\| + \int (|y-y'| + \|x-x'\|)\,\mathrm{d}\gamma(\mu,\nu)\right) \tag{33}$$

Therefore, since the above holds for any coupling $\gamma(\mu,\nu)$ of $\mu\times\nu$, we must have that

$$|\mathcal{E}_\mu^\rho(\theta_\mu) - \mathcal{E}_\nu(\theta_\nu)| \leq S^\rho(\theta_\mu) + \rho\left(\|\theta_\mu - \theta_\nu\| + \inf_{\gamma(\mu,\nu)\in\Gamma(\mu,\nu)}\int (|y-y'| + \|x-x'\|)\,\mathrm{d}\gamma(\mu,\nu)\right) \tag{34}$$

$$= S^\rho(\theta_\mu) + \rho\left(\|\theta_\mu - \theta_\nu\| + W_p(\mu,\nu)\right) \tag{35}$$

Thus, $|\mathcal{E}_\mu^\rho(\theta_\mu) - \mathcal{E}_\nu(\theta_\nu)| \leq S^\rho(\theta_\mu) + \mathcal{O}(\|\theta_\mu - \theta_\nu\| + W_p(\mu,\nu))$, which concludes the proof. $\qquad\square$

## A.2   Sharpness-Aware Domain Adaptation Error

**Theorem 1** (Sharpness-Aware Domain Adaptation Error)**.** Given distributions $\mu,\nu$ over $\mathcal{X}\times\mathcal{Y}$, an error function $\mathcal{E}$ with loss satisfying Assumption 1, the empirical SAM solution $\hat{\theta}_\mu$ in domain $\mu$ satisfying $\mathcal{E}_\mu(\hat{\theta}_\mu) \leq \mathbb{E}_{\epsilon\sim\mathcal{N}(0,\rho^2 I)}[\mathcal{E}_\mu(\hat{\theta}_\mu + \epsilon)]$ for some $\rho > 0$ [6], the population robust risk minimizer $\theta_\nu = \arg\min_{\theta\in\Theta}\mathcal{E}_\nu^\rho(\theta)$ in domain $\nu$, and the population robust risk minimizer $\theta_\mu = \arg\min_{\theta\in\Theta}\mathcal{E}_\mu^\rho(\theta)$ in domain $\mu$, then w.p. $\geq 1-\delta$,

$$\mathcal{E}_\mu(\hat{\theta}_\mu) \leq \mathcal{E}_\nu(\theta_\nu) + \mathcal{O}\left(\frac{\sqrt{k\ln(\|\theta_\mu\|_2^2/\rho^2) + \ln(n/\delta)}}{\sqrt{n}} + \|\theta_\mu - \theta_\nu\| + W_p(\mu,\nu)\right) + S^\rho(\theta_\mu)$$

*Proof.* Let $\theta_\mu = \arg\min_{\theta\in\Theta}\mathcal{E}_\mu^\rho(\theta)$ be the population robust risk minimizer for domain $\mu$ and let $\hat{\theta}_\mu = \arg\min_{\theta\in\Theta}\hat{\mathcal{E}}_\mu^\rho(\theta)$ be the empirical SAM solution for domain $\mu$. We are able to obtain the desired result by applying the sharpness-aware generalization bound in Lemma 2, leveraging the fact that $\hat{\theta}$ is the empirical SAM solution for domain $\mu$, applying the Rademacher complexity generalization bound in Lemma 4, re-

---

[6]This assumption means that adding Gaussian perturbation around $\hat{\theta}_\mu$ increases the expected loss, which should hold for local minima (Foret et al., 2021).

organizing terms, and then applying the robust error difference in Lemma 1:

$$\mathcal{E}_\mu(\hat{\theta}_\mu) \le \hat{\mathcal{E}}_\mu^\rho(\hat{\theta}_\mu) + \mathcal{O}\left(\sqrt{\frac{k\ln(\|\hat{\theta}_\mu\|_2^2/\rho^2) + \ln(n/\delta)}{n}}\right) \qquad \text{(Lemma 2)}$$

$$\le \hat{\mathcal{E}}_\mu^\rho(\theta_\mu) + \mathcal{O}\left(\sqrt{\frac{k\ln(\|\theta_\mu\|_2^2/\rho^2) + \ln(n/\delta)}{n}}\right) \qquad (\hat{\theta} \text{ is the empirical SAM solution for domain } \mu)$$

$$\le \mathcal{E}_\mu^\rho(\theta_\mu) + \mathcal{O}\left(\sqrt{\frac{k\ln(\|\theta_\mu\|_2^2/\rho^2) + \ln(n/\delta)}{n}}\right) + \mathcal{O}\left(\frac{\sqrt{\ln(1/\delta)}}{\sqrt{n}}\right) \qquad \text{(Lemma 4)}$$

$$= \mathcal{E}_\mu^\rho(\theta_\mu) + \mathcal{O}\left(\frac{\sqrt{k\ln(\|\theta_\mu\|_2^2/\rho^2) + \ln(n/\delta)}}{\sqrt{n}}\right)$$

$$\le \mathcal{E}_\nu(\theta_\nu) + \mathcal{O}\left(\frac{\sqrt{k\ln(\|\theta_\mu\|_2^2/\rho^2) + \ln(n/\delta)}}{\sqrt{n}} + \|\theta_\mu - \theta_\nu\| + W_p(\mu,\nu)\right) + S^\rho(\theta_\mu) \qquad \text{(Lemma 1)}$$

$\square$

## A.3 Sharpness-Aware Generalization Bound

**Lemma 2** (Sharpness-Aware Generalization Bound). For any model $\theta \in \Theta \subset \mathbb{R}^k$ satisfying $\mathcal{E}(\theta) \le \mathbb{E}_{\epsilon \sim \mathcal{N}(0,\rho^2 I)}[\mathcal{E}(\theta + \epsilon)]$ for some $\rho > 0$, then w.p. $\ge 1 - \delta$,

$$\mathcal{E}(\theta) \le \hat{\mathcal{E}}^\rho(\theta) + \mathcal{O}\left(\sqrt{\frac{k\ln(\|\theta\|_2^2/\rho^2) + \ln(n/\delta)}{n}}\right) \tag{21}$$

*Proof.* The proof can be found in the appendix of Foret et al. (2021). $\square$

## A.4 Total Sharpness-Aware Error Under GDA

**Theorem 2** (Total Sharpness-Aware Error Under GDA). For any $\delta \in (0,1)$, w.p. $\ge 1 - \delta$, the population risk of the gradually adapted model $\theta_T$ constructed from intermediate models $\theta_0, \ldots, \theta_{T-1}$ according to Algorithm 2, and satisfying $\forall t \in [T]$, $\mathcal{E}(\theta) \le \mathbb{E}_{\epsilon \sim \mathcal{N}(0,\rho_t^2 I)}[\mathcal{E}(\theta_t + \epsilon)]$ for some $\rho_t > 0$[7], can be bounded according to:

$$\mathcal{E}_T(\theta_T) \le \mathcal{E}_0(\theta_0) + \mathcal{O}\left(T(\Delta + \theta_{\text{avg}} + S_{\text{avg}}) + \frac{T(W_{\text{avg}} + \sqrt{\ln(n/\delta)})}{\sqrt{n}} + \sqrt{\frac{\ln(1/\delta)}{nT}} + \sqrt{\ln(nT)}\, \mathcal{R}_{nT}^{\text{seq}}(\Theta)\right)$$

where $\theta_{\text{avg}} := \frac{1}{T}\sum_{t=1}^T \|\theta_t - \theta_{t-1}\|_2$ is the average weight shift, $S_{\text{avg}} := \frac{1}{T}\sum_{t=1}^T S^{\rho_t}(\theta_t)$ is the average sharpness, and $W_{\text{avg}} := \frac{1}{T}\sum_{t=1}^T \mathcal{O}\left(\sqrt{k\ln(\|\theta_t\|_2^2/\rho^2)}\right)$ is the average weight norm.

---

[7]This assumption means that adding Gaussian perturbation to each solution $\theta_t$ increases the expected loss, which should hold for all $\theta_t$ obtained from executing SAM in Algorithm 2 (Foret et al., 2021)

*Proof.* Following the same first two steps as Theorem 1 of Wang et al. (2022), we have:

$$\mathcal{E}_T(\theta_T) \le \sum_{t=0}^{T}\sum_{i=0}^{n-1} q_{nt+i}\mathcal{E}_t(\theta_T) + \text{disc}(\mathbf{q}_{n(T+1)}) + \|\mathbf{q}_{n(T+1)}\|_2 \cdot \mathcal{O}\left(1 + \sqrt{\ln\frac{1}{\delta}}\right)$$
$$+ \mathcal{O}\left(\sqrt{\ln(nT)}\ \mathcal{R}_{nT}^{\text{seq}}(\Theta)\right) \tag{36}$$

$$\le \underbrace{\frac{1}{T+1}\sum_{t=0}^{T}\mathcal{E}_t(\theta_T)}_{(i)} + \underbrace{\text{disc}(\mathbf{q}_{n(T+1)})}_{(ii)} + \mathcal{O}\left(\frac{1}{\sqrt{nT}} + \sqrt{\frac{\ln(1/\delta)}{nT}}\right) + \mathcal{O}(\sqrt{\ln(nT)}\ \mathcal{R}_{nT}^{\text{seq}}(\Theta)) \tag{37}$$

$$\le \mathcal{E}_0(\theta_0) + \mathcal{O}\left(T(\Delta + \theta_{\text{avg}} + S_{\text{avg}}) + \frac{1}{\sqrt{nT}} + \sqrt{\frac{\ln(1/\delta)}{nT}} + \sqrt{\ln(nT)}\ \mathcal{R}_{nT}^{\text{seq}}(\Theta)\right)$$
$$+ \mathcal{O}\left(\frac{T(W_{\text{avg}} + \sqrt{\ln(n/\delta)})}{\sqrt{n}}\right) \tag{38}$$

Then, the last inequality follows from bounding the terms on the second line in the following way:

(i.) To bound term (i.), we can first bound $\mathcal{E}_T(\theta_T)$ by repeatedly applying Theorem 1 in the following way:

$$\mathcal{E}_T(\theta_T) \le \mathcal{E}_{T-1}(\theta_{T-1}) + \mathcal{O}\left(\frac{\sqrt{p\ln(\|\theta_T\|_2^2/\rho_T^2) + \ln(n/\delta)}}{\sqrt{n}} + \|\theta_T - \theta_{T-1}\| + W_p(\mu_T, \mu_{T-1})\right)$$
$$+ S^{\rho_T}(\theta_T) \tag{39}$$

$$\le \mathcal{E}_{T-2}(\theta_{T-2}) + \mathcal{O}\left(\frac{\sqrt{p\ln(\|\theta_T\|_2^2/\rho_T^2) + \ln(n/\delta)}}{\sqrt{n}} + \|\theta_T - \theta_{T-1}\| + W_p(\mu_T, \mu_{T-1})\right)$$
$$+ S^{\rho_T}(\theta_T)$$
$$+ \mathcal{O}\left(\frac{\sqrt{p\ln(\|\theta_{T-1}\|_2^2/\rho_{T-1}^2) + \ln(n/\delta)}}{\sqrt{n}} + \|\theta_{T-1} - \theta_{T-2}\| + W_p(\mu_{T-1}, \mu_{T-2})\right)$$
$$+ S^{\rho_{T-1}}(\theta_{T-1}) \tag{40}$$

$$\le \dots$$

$$\le \mathcal{E}_0(\theta_0) + \sum_{t=1}^{T}(\mathcal{O}\left(\frac{\sqrt{p\ln(\|\theta_t\|_2^2/\rho_t^2) + \ln(n/\delta)}}{\sqrt{n}}\right) + S^{\rho_t}(\theta_t)$$
$$+ \mathcal{O}\left(\|\theta_t - \theta_{t-1}\| + W_p(\mu_t, \mu_{t-1})\right)) \tag{41}$$

Then, the term $\mathcal{E}_{T-1}(\theta_T)$ can be bounded by first applying Lemma 3 and then applying the above bound on $\mathcal{E}_T(\theta_T)$:

$$\mathcal{E}_{T-1}(\theta_T) \le \mathcal{E}_T(\theta_T) + \mathcal{O}(W_p(\mu_T, \mu_{T-1})) \tag{42}$$

$$\le \mathcal{E}_0(\theta_0) + \sum_{t=1}^{T}\left(\mathcal{O}\left(\frac{\sqrt{p\ln(\|\theta_t\|_2^2/\rho_t^2) + \ln(n/\delta)}}{\sqrt{n}}\right) + S^{\rho_t}(\theta_t) + \mathcal{O}(\|\theta_t - \theta_{t-1}\|)\right)$$
$$+ \mathcal{O}\left(W_p(\mu_t, \mu_{t-1}) + W_p(\mu_T, \mu_{T-1})\right) \tag{43}$$

All the successive terms $\mathcal{E}_{T-2}(\theta_T), \mathcal{E}_{T-3}(\theta_T), \dots, \mathcal{E}_0(\theta_T)$ can be bounded the same way. Combining these bounds; defining $\theta_{\text{avg}} := \frac{1}{T}\sum_{t=1}^{T}\|\theta_t - \theta_{t-1}\|_2$ as the average weight shift, $S_{\text{avg}} := \frac{1}{T}\sum_{t=1}^{T}S^{\rho_t}(\theta_t)$ as the average sharpness, and $W_{\text{avg}} := \frac{1}{T}\sum_{t=1}^{T}\mathcal{O}\left(\sqrt{p\ln(\|\theta_t\|_2^2/\rho^2)}\right)$ as the average

weight norm; and refining the upper bound of $\sum_{t=1}^{T} \mathcal{O}\left(\frac{\sqrt{p\ln(\|\theta_t\|_2^2/\rho_t^2)+\ln(n/\delta)}}{\sqrt{n}}\right)$ in the following way

$$\sum_{t=1}^{T} \mathcal{O}\left(\frac{\sqrt{p\ln(\|\theta_t\|_2^2/\rho_t^2)+\ln(n/\delta)}}{\sqrt{n}}\right) = \sum_{t=1}^{T} \mathcal{O}\left(\sqrt{p\ln(\|\theta_t\|_2^2/\rho_t^2)+\ln(n/\delta)}\right) + \mathcal{O}\left(\frac{T}{\sqrt{n}}\right) \qquad (44)$$

$$\leq \sum_{t=1}^{T} \mathcal{O}\left(\sqrt{p\ln(\|\theta_t\|_2^2/\rho_t^2)} + \sqrt{\ln(n/\delta)}\right) + \mathcal{O}\left(\frac{T}{\sqrt{n}}\right) \qquad (45)$$

$$\leq \sum_{t=1}^{T} \mathcal{O}\left(\sqrt{p\ln(\|\theta_t\|_2^2/\rho_t^2)}\right) + \mathcal{O}\left(\frac{T(\sqrt{\ln(n/\delta)})}{\sqrt{n}}\right) \qquad (46)$$

$$= \mathcal{O}\left(\frac{T(W_{\text{avg}} + \sqrt{\ln(n/\delta)})}{\sqrt{n}}\right) \qquad (47)$$

yields the following bound:

$$\frac{1}{T+1}\sum_{t=0}^{T} \mathcal{E}_T(\theta_T) \leq \mathcal{E}_0(\theta_0) + \mathcal{O}\left(T(S_{\text{avg}} + \theta_{\text{avg}} + \Delta) + \frac{T(W_{\text{avg}} + \sqrt{\ln(n/\delta)})}{\sqrt{n}}\right) \qquad (48)$$

(ii.) To bound term (ii.), we apply Proposition 1 with

$$\mathbf{q}_{n(T+1)} = \mathbf{q}_{n(T+1)}^\star := \left(\frac{1}{(n(T+1)}, \dots, \frac{1}{(n(T+1)}\right) \qquad (49)$$

$\square$

## A.5 Helper Lemmas

**Lemma 3** (Standard Error Between Shifted Domains - Lemma 1 of Wang et al. (2022))**.** Given an error function $\mathcal{E}_\mu(\theta) := \mathbb{E}_{(\mathbf{x},y)\sim\mu}[\ell(y,\mathbf{x},\theta)]$ with loss satisfying Assumption 1 and any measures $\mu, \nu$ on $\mathcal{Y} \times \mathcal{X}$, we have that

$$|\mathcal{E}_\mu(\theta) - \mathcal{E}_\nu(\theta)| \leq \mathcal{O}(W_p(\mu,\nu)) \qquad (50)$$

*Proof.* The proof can be found in the appendix of Wang et al. (2022). $\square$

**Proposition 1** (Discrepancy Bound - Lemma 2 of Wang et al. (2022))**.**

$$\text{disc}(\mathbf{q}_t) \leq \mathcal{O}\left(\sum_{k=0}^{t-1} q_k(t-k-1)W_p(\mu_k,\mu_{k+1})\right) \qquad (51)$$

Furthermore, if we let $\mathbf{q}_t = \mathbf{q}_t^\star := (1/t, \dots, 1/t)$, then

$$\text{disc}(\mathbf{q}_t) \leq \mathcal{O}(t\Delta) \qquad (52)$$

*Proof.* The proof can be found in the appendix of Wang et al. (2022). $\square$

**Definition 8** (Rademacher Complexity)**.** The empirical Rademacher complexity of a model class $\Theta \subset \mathbb{R}^d$ with induced classifiers $f_\theta, \theta \in \Theta$ on a set of i.i.d. samples $S := \{x_i, \dots, x_n\} \sim \mu$, where $\mu \in \Delta(\mathbb{R}^d)$, is given by

$$\mathcal{R}_\mu(\Theta) := \frac{1}{n}\mathbb{E}_{\epsilon\sim\{\pm1\}^n}\left[\sup_{\theta\in\Theta}\sum_{i=1}^{n}\epsilon_i f_\theta(x_i)\right] \qquad (53)$$

Following Wang et al. (2022), we assume the Rademacher Complexity of our model family is bounded for all distributions $\mu \in \Delta(\mathbb{R}^d)$.

**Assumption 2** (Bounded Rademacher Complexity)**.** There exists some $B > 0$ so that for any set of $n$ samples drawn i.i.d. from $\mu \in \Delta(\mathbb{R}^d)$, we have that $\mathcal{R}_\mu(\Theta) \leq \frac{B}{\sqrt{n}}$.

**Lemma 4** (Rademacher Complexity Generalization Bound)**.** If Assumption 2 holds, then for any $\theta \in \Theta$, the absolute difference between its empirical and population error can be upper bounded according to:

$$|\mathcal{E}(\theta) - \hat{\mathcal{E}}(\theta)| \leq \mathcal{O}\left( \frac{\sqrt{\ln(1/\delta)}}{\sqrt{n}} \right) \tag{54}$$

*Proof.* See Lemma A.1 of Kumar et al. (2020). □

## B  SAM Variant Hyperparameters

Our experiments and hyperparameter search are performed over three random seeds for varying values of $\rho \in \{0.01, 0.02, 0.05, 0.1, 0.2\}$, with the exception of ASAM. For ASAM, the authors suggest using a value of $\rho$ that is roughly 10 times larger than the value for SAM, so we test $\rho \in \{0.1, 0.2, 0.5, 1, 2\}$. For K-SAM, we follow the guidance from the original paper and set $K = B/2$, where $B$ is the batch size. For LookSAM, following the recommendations of the authors, we set $k = 5$ and test $\alpha \in \{0.5, 0.7, 0.1\}$. For FriendlySAM, following the original experiments, we set $\sigma = 1$ and test $\phi \in \{0.6, 0.9, 0.95\}$. For ESAM, following the original experiments, we set $\gamma = 0.5$ and we test $\xi \in \{0.5, 0.6\}$. Finally, for FisherSAM, we test $\eta \in \{0.01, 0.2, 0.5, 1\}$. In Table 1, we report the best accuracy values obtained over all hyperparameter settings for each SAM variant on each of the four datasets.

