# OpenReview forum: "Towards Understanding the Role of Sharpness-Aware Minimization Algorithms for Out-of-Distribution Generalization"
_TMLR — Rejected by TMLR_

### Review · Reviewer_Jo8i · 2025-01-28

**Summary Of Contributions:**

This paper investigates how Sharpness-Aware Minimization (SAM) and its variants enhance model generalization in out-of-distribution (OOD) contexts. The authors conduct a detailed empirical comparison across eight SAM variants, revealing significant improvements in zero-shot OOD generalization, with original SAM outperforming the Adam baseline by 4.76% and the best variants achieving an 8.01% improvement. Extending the analysis to gradual domain adaptation (GDA), the study demonstrates that SAM consistently boosts accuracy over Adam, particularly with variants like FriendlySAM and FisherSAM. Theoretical contributions include novel sharpness-based generalization bounds for both OOD and GDA settings, offering insights into the relationship between sharpness, distribution shifts, and model performance. The authors also identify a gap between theoretical guarantees and empirical results, suggesting future work to develop tighter bounds that account for SAM-specific characteristics and ultimately reconcile theory with observed practical gains.

**Audience:**

Yes

**Claims And Evidence:**

Yes

**Requested Changes:**

1. Provide more detailed and nuanced discussions on the theoretical and practical implications of SAM in the broader scope of OOD generalization beyond GDA.

2. Correct the repeated phrase on page 1: "SAM remains remains...".

3. Conduct a thorough proofreading pass to identify and resolve any other minor grammatical or typographical errors.

4. Provide clearer guidelines for practitioners on selecting SAM variants and tuning hyperparameters (e.g., perturbation radius
𝜌) based on specific OOD scenarios.

**Strengths And Weaknesses:**

### Strengths
1. The paper provides a thorough comparison of eight SAM variants in zero-shot OOD generalization and gradual domain adaptation (GDA) across four diverse benchmarks.

2. The authors derive sharpness-aware generalization bounds for OOD settings and extend these to the GDA framework.

3. The paper identifies key gaps in theoretical bounds and proposes actionable future directions, such as localized analyses that leverage SAM's implicit properties like denoising and feature regularization.

### Weaknesses

1. The derived generalization bounds, while insightful, are asymptotically similar to prior work and fail to fully capture the strong empirical performance of SAM, particularly in GDA.

2. The paper does not propose a new algorithm or methodological improvement for enhancing OOD generalization, focusing instead on analyzing and benchmarking existing SAM variants.

3. The study primarily focuses on Gradual Domain Adaptation (GDA), which is a relatively narrow and specific topic. A broader exploration of the theoretical framework to encompass other OOD generalization scenarios would significantly enhance the paper’s impact and relevance.

---

> ### Author Response · Authors · 2025-03-13
> **Rebuttal by Authors**
>
> We thank the reviewer for the constructive comments on helping improve our work.
>
> > RC1: Provide more detailed and nuanced discussions on the theoretical and practical implications of SAM in the broader scope of OOD generalization beyond GDA.
>
> A: Thank you for suggesting this change. In the related works section, under the category “Sharpness and OOD generalization,” we have added additional commentary on the role of SAM in OOD generalization for increasingly popular large language model applications, including cross-lingual transfer and multi-task generalization.
>
> > RC2: Correct the repeated phrase on page 1: "SAM remains remains...".
>
> > RC3: Conduct a thorough proofreading pass to identify and resolve any other minor grammatical or typographical errors.
>
> A: Thank you for pointing this out. We have fixed the repeated phrase on page 1 and conducted a proofreading pass to fix any other grammar errors.
>
> > RC4: Provide clearer guidelines for practitioners on selecting SAM variants and tuning hyperparameters (e.g., perturbation radius 𝜌) based on specific OOD scenarios.
>
> Thank you for your comment. In the appendix of the paper, we have provided detailed guidelines on hyperparameters used for all the SAM variants, including all the values we used in our grid search. However, there might not be a universal law for choosing the various hyperparameters across SAM variants since the optimal values are largely dataset dependent and practitioners would have to perform the same grid search that we did.

---

> > ### Comment · Reviewer_Jo8i · 2025-03-30
> >
> > Thank you for the response. My requested changes have been made, but the main weaknesses are still there.

---

### Review · Reviewer_3DdX · 2025-02-17

**Summary Of Contributions:**

This paper makes contributions to the understanding of Sharpness-Aware
Minimization (SAM) algorithms in the context of Out-of-Distribution (OOD) generalization.
The contributions can be divided into two parts: those explicitly stated by the authors and
additional insights stated by the reviewers derived from this paper. The key contributions of
this paper can be summarized as follows

- A comprehensive comparison of eight SAM variants in the context of zero-shot OOD generalization. Through extensive experiments, the authors demonstrate that the original SAM outperforms the Adam
baseline by 4.76%, while the strongest SAM variants achieve an 8.01%
improvement over Adam on average.

- In addition to the empirical evaluation, the paper provides a theoretical OOD
generalization bound in terms of sharpness. By deriving a novel bound that explicitly
links sharpness to OOD generalization, the authors oﬀer a formal explanation of
how sharpness-aware optimization influences model robustness.

- This paper extends the study of SAM to Gradual
Domain Adaptation (GDA), a more structured form of OOD generalization where
intermediate domains are introduced between the source and target distributions. In
this setting, the model is iteratively trained using self-training on unlabeled
intermediate domains to improve its adaptability to the target domain. The
experimental results indicate that the original SAM outperforms the Adam baseline
by 0.82% on average, while the strongest SAM variants achieve a 1.52%
improvement over Adam, reinforcing the eﬀectiveness of sharpness-aware training
in GDA scenarios.

**Audience:**

Yes

**Claims And Evidence:**

Yes

**Requested Changes:**

In general, the paper is well-structured. I do not have major issues. See the weakness parts for the minor issues.

**Strengths And Weaknesses:**

**Summary** This paper presents a thorough empirical and theoretical study of SAM algorithms in the
context of OOD generalization and Gradual Domain Adaptation. While the paper has
several strong aspects that contribute to a deeper understanding of sharpness-aware
optimization in domain shift scenarios, there are also some weaknesses that may require
further refinement. Below, I provide a detailed analysis of both the strengths and
weaknesses of this submission.

**Strengths**

-  One of the key strengths of this paper is its comprehensive evaluation of multiple
SAM variants, not only in terms of their performance but also in terms of their
computational costs on page 5. The authors systematically compare eight diﬀerent
SAM models (including the original SAM), providing a clear analysis of their
computational costs. This comparison allows readers to understand the details of
these eight diﬀerent SAM models when selecting a sharpness-aware optimization
method.

- The rigorous mathematical derivation of key theoretical results.

- A well-structured description of diﬀerent SAM models. This is beneficial for readers (like me) who
may not be familiar with the nuances of various sharpness-aware minimization techniques.

- This paper addresses an important gap in the study of SAM for OOD generalization, an area that has not been extensively explored in previous research. While SAM has been widely studied for improving iid generalization, its eﬀectiveness in OOD settings and gradual domain adaptation was previously
unclear.

**Weakness**

- Inconsistencies between diﬀerent sections. For example, in Section 3.1 (page 6), the paper provides a detailed description of how the datasets are divided, including the number of samples in the source
domain and intermediate domains. The numbers presented in this section suggest that the source and intermediate domains contain comparable amounts of data. However, in Section 4.1 (page 8), the paper states that the unlabeled training examples in the intermediate domains are significantly fewer than the labeled training examples in the source domain.

- The paper is seemingly relied on ideas from Wen et al. (2023) without suﬃcient diﬀerentiation. In Section 3.3 (page 7), the paper builds upon some of the theoretical insights from Wen et al. (2023) and aims to refine them, but Lemma 1 appears to be largely derived from Wen et al. (2023).
Furthermore, Lemma 2’s formulation resembles a generalization bound for an i.i.d.
setting, as the ordering of domains does not appear to impact the final result. Because Lemma 2 is based on an i.i.d. setting and Lemma 1 is derived from prior work, it naturally leads to the conclusion that “asymptotically, this generalization bound is no better than the one for self-training in the literature of GDA” from the Abstract.

-  The approach used for Gradual Domain Adaptation in this paper is relatively
simple. The authors implement GDA by treating it as a sequence of n+1 zero-shot
OOD generalization tasks, where each target domain is simply the next intermediate
domain in the sequence until reaching the final target domain. While this approach
provides a natural extension of zero-shot OOD generalization, it does not fully
leverage modern sequential domain adaptation techniques or continual learning tech that could have
enhanced the eﬀectiveness of SAM in GDA. This simplification may partially explain
why SAM’s improvement in the GDA setting is relatively small compared to its
performance in zero-shot OOD generalization.

---

> ### Author Response · Authors · 2025-03-13
> **Rebuttal by Authors**
>
> We thank the reviewer for the constructive comments on helping improve our work.
>
> > W1: Inconsistencies between diﬀerent sections. For example, in Section 3.1 (page 6), the paper provides a detailed description of how the datasets are divided, including the number of samples in the source domain and intermediate domains. The numbers presented in this section suggest that the source and intermediate domains contain comparable amounts of data. However, in Section 4.1 (page 8), the paper states that the unlabeled training examples in the intermediate domains are significantly fewer than the labeled training examples in the source domain.
>
> A: Thank you for pointing this out. We originally made this claim in Section 4.1 because it tends to be quite common in practical applications beyond those tested in our paper, but since this assumption contradicts the experimental setup and, most importantly, is not needed at all for our proof, we have updated the paper to remove this assumption and avoid any potential confusion.
>
> > W2: The paper is seemingly relied on ideas from Wen et al. (2023) without suﬃcient diﬀerentiation. In Section 3.3 (page 7), the paper builds upon some of the theoretical insights from Wen et al. (2023) and aims to refine them, but Lemma 1 appears to be largely derived from Wen et al. (2023). Furthermore, Lemma 2’s formulation resembles a generalization bound for an i.i.d. setting, as the ordering of domains does not appear to impact the final result. Because Lemma 2 is based on an i.i.d. setting and Lemma 1 is derived from prior work, it naturally leads to the conclusion that “asymptotically, this generalization bound is no better than the one for self-training in the literature of GDA” from the Abstract.
>
> A: Thank you for the comment. We believe you might have made a minor typo in this response and meant to refer to Wang et al. (2022) wherever you mention Wen et al. (2023). Regarding Lemma 1, our work does extend Wang et al. (2022), but Lemma 1 is considering the sharpness-aware loss in the first term of the LHS, so it is different from Wang et al. (2022), which considers the standard loss. Although it uses the same proof technique as Wang et al. (2022), it is not derived directly from the prior work. Regarding Lemma 2, which is based on the i.i.d. setting, this is simply stated as a building block for the final theorem, so that when Lemma 2 and Lemma 1 are used together, Theorem 1 can be obtained. Theorem 1 directly considers the OOD setting.
>
> > W3: The approach used for Gradual Domain Adaptation in this paper is relatively simple. The authors implement GDA by treating it as a sequence of n+1 zero-shot OOD generalization tasks, where each target domain is simply the next intermediate domain in the sequence until reaching the final target domain. While this approach provides a natural extension of zero-shot OOD generalization, it does not fully leverage modern sequential domain adaptation techniques or continual learning tech that could have enhanced the eﬀectiveness of SAM in GDA. This simplification may partially explain why SAM’s improvement in the GDA setting is relatively small compared to its performance in zero-shot OOD generalization.
>
> A: Thank you for the insightful remark. First, we want to emphasize that in GDA, the sequence of domains is given to the learner sequentially. Therefore, at each time step, the next intermediate domain should indeed be treated as a target domain. Second, we completely agree that coupling SAM with more sophisticated domain adaptation/continual learning techniques could lead to even further improvements for GDA and is worth investigating. However, we feel that layering SAM on top of more sophisticated techniques is tangential to the aim of our study, which is to understand the impact of SAM on GDA directly.

---

> > ### Comment · Reviewer_3DdX · 2025-03-23
> >
> > Thank you. My concerns have been addressed.

---

### Review · Reviewer_PrUE · 2025-02-28

**Summary Of Contributions:**

In this work, the authors study the family of sharpness aware minimization methods in the context of distribution shifts. Previous works largely focused on SAM based methods and their benefits in standard i.i.d. generalization setting. The work is divided into two parts. In the first part, the authors study standard zero shot generalization problems and in the second part the authors study gradual domain adaptation problems. In the first part, the authors provide a new generalization bound that connects error of target domain to source domain in terms of Wasserstein distance between the distributions and rho-sharpness. The authors conduct experiments to show that variants of SAM are better than Adam in zero shot generalization settings. In the second part, the authors extend their theoretical analysis to gradual domain adaptation settings. In this part, the authors use SAM with self-training based approach. The authors adapt their results for self-training in gradual domain adaptation. They also present experiments and show that SAM based approaches turn out to still be advantageous when compared with Adam.

**Audience:**

Yes

**Broader Impact Concerns:**

I cannot foresee any ethical concerns.

**Claims And Evidence:**

No

**Requested Changes:**

Please see the weaknesses section.

**Strengths And Weaknesses:**

Strengths

i) Sharpness aware minimization algorithms have become popular and present a promising alternate to more conventional SGD and Adam based approaches. Despite their popularity, a systematic analysis (theoretical and empirical) of SAM has been missing in the setting beyond i.i.d. generalization. Hence, the key theme and questions investigated in this paper is important.

ii) From the point of view of coverage, the authors have considered an extensive set of variants of SAM, which is notable and appreciated.

Weaknesses

i) In Section 3, the authors present an analysis on zero shot generalization to compare SAM with Adam. The set of datasets considered are far from standard and complete for out of distribution generalization literature. https://wilds.stanford.edu/, https://github.com/facebookresearch/DomainBed, https://wild-time.github.io/, https://github.com/MadryLab/BREEDS-Benchmarks, https://github.com/YyzHarry/SubpopBench. The authors consider colored MNIST, rotated MNIST and a rather small potraits dataset for vision tasks. The results from these datasets provide a very biased signal. If we go to more complex datasets, we may find that SAM is not that helpful afterall.

ii) The comparisons with Adam seem unfair. It seems authors have optimized the hyperparameters rho for the SAM based methods but it seems default hyperparameters are used. Domain generalization approaches have been known to be very sensitive to type of hyperparameter optimization. Please see https://github.com/facebookresearch/DomainBed.

iii) I have concerns about theoretical results as well. In Theorem 1, the authors show that

$$E_{\mu} (\theta_{\mu}) \leq E_{\nu}(\theta_{\nu}) + S^{\rho}(\theta_{\mu}) + O(\||\theta_{\mu}-\theta_{\nu}\|| + W_{p}(\nu, \mu)) + O(c/\sqrt{n})$$

The bound seems to be loose and can be made tighter. We do not need to combine Lemma 1 and Lemma 2. If we consider the proof steps in Lemma 1 (equation 30 to equation 37), we obtain

$$E_{\mu} (\theta_{\mu}) \leq E_{\nu}(\theta_{\nu}) + O(\||\theta_{\mu}-\theta_{\nu}\|| + W_{p}(\nu, \mu)) $$

Thus there is no need for the term  $S^{\rho}(\theta_{\mu})$ and $O(c/\sqrt{n})$. Basically, we only need to consider the second term in the RHS in (equation 30 to equation 35) to arrive at the above.
Hence, the above bound is tighter and any dependence on sharpness parameter makes it looser.  Thus, the entire Theorem 1 seems to be incorrect and present a trivially vacuous claim.

iv) The other theoretical result seems to be a corollary of Theorem 2 from Wang et al. The bound seems to not depend on any sharpness related properties and is same as bound in Wang et al. I do not quite see the point of this result as it does not introduce anything new.

---

> ### Author Response · Authors · 2025-03-13
> **Rebuttal by Authors**
>
> We thank the reviewer for the constructive comments on helping improve our work.
>
> > W1: In Section 3, the authors present an analysis on zero shot generalization to compare SAM with Adam. The set of datasets considered are far from standard and complete for out of distribution generalization literature. https://wilds.stanford.edu/, https://github.com/facebookresearch/DomainBed, https://wild-time.github.io/, https://github.com/MadryLab/BREEDS-Benchmarks, https://github.com/YyzHarry/SubpopBench. The authors consider colored MNIST, rotated MNIST and a rather small potraits dataset for vision tasks. The results from these datasets provide a very biased signal. If we go to more complex datasets, we may find that SAM is not that helpful afterall.
>
> A: Thank you for the comment. We are running an additional set of experiments on more complex datasets from DomainBed. We will post a final version of the draft with the additional experimental results as soon as they are available.
>
> > W2: The comparisons with Adam seem unfair. It seems authors have optimized the hyperparameters rho for the SAM based methods but it seems default hyperparameters are used. Domain generalization approaches have been known to be very sensitive to type of hyperparameter optimization. Please see https://github.com/facebookresearch/DomainBed.
>
> A: Thank you for the comment. We first want to clarify that all of our SAM variants use the Adam method as the base optimizer, so that SAM with rho=0.00 is exactly equivalent to Adam. Then, to isolate the effect of the perturbation radius, for SAM, we performed the ablation over rhos=[0.01, 0.02, 0.05, 0.1, 0.2] directly on top of the Adam optimizer. Therefore, the comparison of SAM versus Adam simply explores how increasing the perturbation radius on top of Adam affects the overall performance. For the SAM variants with additional hyperparameters, we do perform a grid search over the additional hyperparameters, but we again emphasize that the base optimizer is always Adam, so we are simply making the SAM-vvariant modifications on top of the same base Adam optimizer. Therefore, we feel the comparison is fair and reasonable.
>
> > W3: I have concerns about theoretical results as well...
>
> A: Thank you for the insightful comments and careful review of our work. We intended to give a generalization bound on the learned classifier in Theorem 1 by using the empirically obtained SAM solution $\hat{\theta}_\mu$ on the LHS of the inequality, in which case it is indeed necessary to combine Lemma 1 and Lemma 2, and we have revised our theorem statement and proof accordingly to address this.
>
> Now, considering the revised version of Theorem 1, the dependency on the sharpness term, $S^\rho(\theta_\mu)$, is still needed in Lemma 1, since in Theorem 1 we are considering the solution obtained from the SAM model directly, which aims to minimize the \rho-robust risk. Moreover, the $O(c/\sqrt{n})$ term resulting from Lemma 2 is also needed in Theorem 1, since the learner needs access to the dataset in order to get the learned model $\hat{\theta}_\mu$. We have modified the theorem statement and proof of Theorem 1 to reflect this change.
>
> The revised version of Theorem 1 also now gives a comparison of the learned model in domain $\mu$ with respect to the optimal model performance from the target domain $\nu$.
>
> > W4: The other theoretical result seems to be a corollary of Theorem 2 from Wang et al. The bound seems to not depend on any sharpness related properties and is same as bound in Wang et al. I do not quite see the point of this result as it does not introduce anything new.
>
> A: Thank you for the comment. Even though our results do not provide an asymptotically tighter bound, we feel the discussion in Section 4.6 where we explain potential avenues for how to obtain sharper bounds in the future can provide a valuable starting point for others working to explore the theoretical benefit of SAM for OOD generalization, expediting the rate of progress in this area and consolidating the present challenges into one framework.

---

> > ### Comment · Reviewer_PrUE · 2025-03-30
> > **Response**
> >
> > Thanks for the clarifications.  I continue to stand with the view that the set of datasets used are far from standard and complete for out of distribution generalization literature. I recommend that the authors add some datasets from DomainBed and some datasets from WILDS to have a much better coverage.

---

### Author Response · Authors · 2025-03-13
**Author Comment and Revision Summary**

We thank all the three reviewers for the detailed comments and constructive feedback. We have uploaded a revised version of the draft with the reviewer feedback incorporated.

The main concern on the theoretical results is that the theory does not provide new mathematical insight into the model for the OOD setting. Instead, the theory simply extends Foret et al. (2021) to the setting of Wang et al. (2022), without any substantial novelty introduced.

In general, our rebuttal to all such concerns (R2 W2, R3 W4) is the following:
- Our theoretical results analyze the effect of SAM under OOD settings, including both domain generalization and gradual domain adaptation, and are the first of their kind.
- Combining these frameworks into an OOD analysis for SAM culminates in a valuable discussion in Section 4.6 where we explain, very clearly, potential avenues for how to obtain sharper bounds in the future. The discussion is grounded in the rigorous exploration of the two theoretical frameworks.
- Even if our results do not provide an asymptotically tighter bound, this discussion and the mathematical exploration can provide a valuable starting point for others working to explore the theoretical benefit of SAM for OOD generalization, expediting the rate of progress in this area and consolidating the present challenges into one framework.

---

### Decision · Action_Editor_fzbd · 2025-04-15

**Recommendation:** Reject

**Comment:**

Please check my comments in "Claims And Evidence".

I would recommend a major revision that includes:

1. Comprehensive empirical evaluations following the DomainBed protocol (including hyperparameter search and model selection)
2. Comprehensive comparisons with the existing DG methods -- solely using SAM might not achieve the "state-of-the-art". One possible option might be using SAM as an optimizer of the existing methods. For example, many papers show state-of-the-art performances using SWAD as their optimizer.
3. Deeper theoretical analysis and comparison with the existing "sharpness vs. DG" studies. For example, Cha et al 2021 and Zou et al 2024 have theoretically shown the connection between robustness and sharpness because the relationship between robustness and sharpness itself is not specifically novel.

**Audience:**

Both sharpness and OOD generalization are popular topics in the field. There might be some audiences interested in the intersection between them.

**Claims And Evidence:**

This paper studies Sharpness-Aware Minimization (SAM) for out-of-distribution (OOD) generalization or domain generalization (DG). This paper shows that SAM variants significantly outperform Adam in both zero-shot OOD and gradual domain adaptation settings. The authors provide sharpness-based generalization bounds, though these don’t fully explain SAM’s empirical success, highlighting directions for future theoretical work.

The main claim is that SAM is helpful for DG and GDA empirically and theoretically. The authors supported this claim by (1) empirically providing experiments on four datasets (Rotated MNIST Color MNIST Covertype Portraits) with eight SAM variants and the Adam baseline, and (2) theoretically showing that SAM is helpful for the domain generalization error.

However, as pointed out by the reviewers, the empirical support is somewhat weak.

1. Limited benchmarks: The chosen benchmarks are far from standard DG benchmarks (e.g., DomainBed),
2. Limited comparison methods: DG is a popular research field that has many previous studies.

For point 1, I think that comparing methods on a specific benchmark (especially for very small-scale datasets) could be misleading. This is because as shown by the DomainBed paper, (1) DG methods are highly sensitive to hyperparameters; we need a unified hyperparameter tuning setting as suggested by DomainBed, and (2) the performance could vary by the choice of the datasets. The authors mentioned that they would add the DomainBed experiments in the revision, but I couldn't find the results in the revision.

For point 2, just showing that SAM is better than Adam (where the standard optimization choice is the SGD, or SWAD) would not be sufficient support that SAM is really helpful for DG compared to the previous DG methods. Specifically, as the paper mentioned in "Sharpness and OOD Generalization", the relationship between sharpness and DG has already been studied by Cha et al 2021, Zhang et al 2023, and Zou et al 2024.

Both sharpness and OOD generalization are popular topics in the field. However, the relationship between sharpness and OOD generalization has already been explored by many previous studies. For example, Cha et al 2021 and Zhang et al 2023 have shown that sharpness-aware optimization, especially for SAM, can perform better than the standard optimization in DomainBed -- both Cha et al and Zhang et al showed the results. This paper only extends this empirical study to other SAM variants on the limited benchmarks.

Furthermore, Cha et al 2021 and Zou et al 2024 have theoretically shown the connection between robustness and sharpness; I think the relationship between them is not specifically novel. Similarly, as pointed out by Reviewer 3DdX and Reviewer Jo8i, the theoretical results of this paper would rely on the previous theoretical results. Although the theoretical result is different from Cha et al 2021, Zou et al 2024 or Wen et al 2022, the relationship between sharpness and robustness itself is not specifically new.

The reviewers raised some concerns regarding the theoretical results, e.g., the results heavily relied on previous works (3DdX, Jo8i) -- The reviewer confirmed that their concerns were resolved after the revision.

**Resubmission Of Major Revision:**

The authors may consider submitting a major revision at a later time.